# Loneliness corresponds with neural representations and language use that deviate from shared cultural perceptions
Timothy W. Broom [1] ✉, Siddhant Iyer[2], Andrea L. Courtney [3] & Meghan L. Meyer [1] ✉

The word zeitgeist refers to common perceptions shared in a given culture. Meanwhile, a defining feature of loneliness is feeling that one's views are not shared with others. Does loneliness correspond with deviating from the zeitgeist? Across two independent brain imaging datasets, lonely participants' neural representations of well-known celebrities strayed from group-consensus neural representations in the medial prefrontal cortex — a region that encodes and retrieves social knowledge (Studies 1 A/1B: $N = 40$ each). Because communication fosters social connection by creating shared reality, we next asked whether lonelier participants' communication about well-known celebrities also deviates from the zeitgeist. Indeed, when a strong group consensus exists, lonelier individuals use idiosyncratic language to describe well-known celebrities (Study 2: $N = 923$). Collectively, results support lonely individuals' feeling that their views are not shared. This suggests loneliness may not only reflect impoverished relationships with specific individuals, but also feelings of disconnection from prevalently shared views of contemporary culture.

In November of 2011, *Jimmy Kimmel Live* featured a skit in which Jimmy Kimmel and Ellen DeGeneres had a "nice off" – a contest of increasingly considerate acts to see who was the nicer of the two TV show hosts. The entire premise of the skit relied on the audience's awareness of the public perception at the time of Ellen DeGeneres as a kind person. The show's writers assumed that their millions of viewers held a common understanding of who Ellen DeGeneres was as a person and that it would be funny to riff on this shared perception precisely because of its ubiquity. The very notion of public perception takes for granted that similar mental representations of celebrities are shared by large portions of the population. A similar idea is captured by the German word zeitgeist – literally translated as "spirit of the time" – which refers to the common perceptions and cognitions shared between members of a given culture. But do we all align our views with the zeitgeist equally? Who watches something like the "nice off" skit and is in on the joke? Conversely, who is left scratching their head or scoffing, and are there any downsides associated with straying from the dominant view?

We propose lonely individuals' mental representations of contemporary culture stray from the zeitgeist, specifically the group-consensus representations of cultural knowledge. This prediction stems from three observations. First, the subjective perception that one's ideas are not shared by others is a defining feature of loneliness[1,2]. It is possible that there is

ground truth to this subjective perception, with lonelier individuals objectively representing contemporary cultural knowledge idiosyncratically. Second, group-consensus representations of prominent celebrities' traits and attributes predict neural responses to those same celebrities in an independent group[3]. In other words, celebrities are represented in a similar way across different individuals with respect to distributed patterns of neural activity. Because celebrities are widely recognized figures in contemporary culture, this outcome implies that the zeitgeist (i.e., shared cultural perceptions) might be reflected in these shared neural representations. Additionally, because these prior results already demonstrated a consensus in neural representations of prominent celebrities, they suggest studying celebrity perception is a reasonable starting place to test whether loneliness is associated with mental representations that differ from prevailing cultural trends.

The third piece of evidence supporting our prediction comes from the growing body of research suggesting social connection between people is mirrored by similarity in their neural responses to popular culture media[4–7]. Individuals who are objectively closer in a social network move through similar psychological states while watching video clips taken from popular media (e.g., a scene from *America's Funniest Home Videos*)[4,5]. Moreover, objective and subjective social isolation predict idiosyncratic neural responding while viewing entertaining video footage[6,7], suggesting lonely

[1]Department of Psychology, Columbia University, New York, NY, USA. [2]Department of Neuroscience, Columbia University, New York, NY, USA. [3]Department of Psychology, Stanford University, Stanford, CA, USA. ✉e-mail: twb2112@columbia.edu; mlm2378@columbia.edu

individuals may process popular media unconventionally. Here we investigate whether lonely individuals demonstrate neural idiosyncrasy relative to the zeitgeist, i.e., widely shared perceptions of contemporary culture, including prominent celebrities who are fixtures of modern popular culture. Filling this gap is important because contemporary cultural knowledge is frequently referenced spontaneously in social life to facilitate social connection[8–10]. For example, celebrities that generate common ground are disproportionately discussed in conversations between strangers[11]. In everyday social life, possessing representations that deviate from the zeitgeist may generate feelings of isolation and/or place a person at a disadvantage when it comes to identifying common ground with others. Loneliness may likewise generate idiosyncratic representations of popular culture knowledge, for example by reducing the motivation to search for common ground with others. Overall, past work hints to the possibility that loneliness is associated with idiosyncratic views of contemporary culture, including perceptions of celebrities.

We investigated our hypothesis that loneliness corresponds with mental representations that deviate from the zeitgeist with two different objective measures of contemporary cultural perceptions: neural representations measured while reflecting on well-known celebrities and linguistic descriptions while communicating about well-known celebrities. In terms of neural representations, we assessed whether lonely participants' multivariate neural patterns of activity while reflecting on well-known celebrities are idiosyncratic relative to less lonely participants. We leveraged two independent fMRI datasets (Study 1 A and 1B) in which all participants completed a trait evaluation task for the same set of five prominent celebrities. In each dataset, spatial patterns of neural activity for each celebrity were directly compared across participants in two conceptually related but distinct tests of our hypothesis. First, we calculated similarity in neural representations of celebrities for every possible pair of participants and

tested for an Anna Karenina effect[12], which derives its name from the opening quote of Tolstoy's famous novel: "All happy families are alike; each unhappy family is unhappy in its own way." Here, we tested whether all socially connected people are alike in their representations of celebrities, while each lonely individual perceives celebrities in their own way. In other words, our Anna Karenina model tested whether lonelier participants were especially dissimilar in their neural representations of celebrities, relative to both other lonely participants and less lonely participants. This first approach tests whether lonely individuals have idiosyncratic representations of well-known celebrities compared to other individuals.

While this first pairwise approach is similar in nature to previous work examining the association between shared neural responses and social connection[4–7,13], we further compared each participant's neural representation of a celebrity to the group-consensus neural representation of that celebrity. Previous studies have used group-average neural responses to identify brain regions that differentiate between two groups[14,15] and to assess the extent to which one's neural representation of self corresponds to their peers' collective neural representation of her or him[16]. In each case, the underlying assumption is that there is a prototypical response that is best captured by averaging individual responses together. In the current study, we aimed to capture not just the overall group response, but the point of greatest convergence across individuals that would best reflect the most commonly held perceptions of each celebrity (much like the concept of the zeitgeist), reasoning that straying from this point of greatest convergence would have the most widespread implications for one's sense of belonging. Therefore, we defined the group-consensus neural representation of a celebrity as the weighted average across all participants with greater weight being given to those participants who were closest to this point of convergence (i.e., the point with the greatest density of surrounding participants, see Fig. 1). This second approach tests whether lonelier individuals

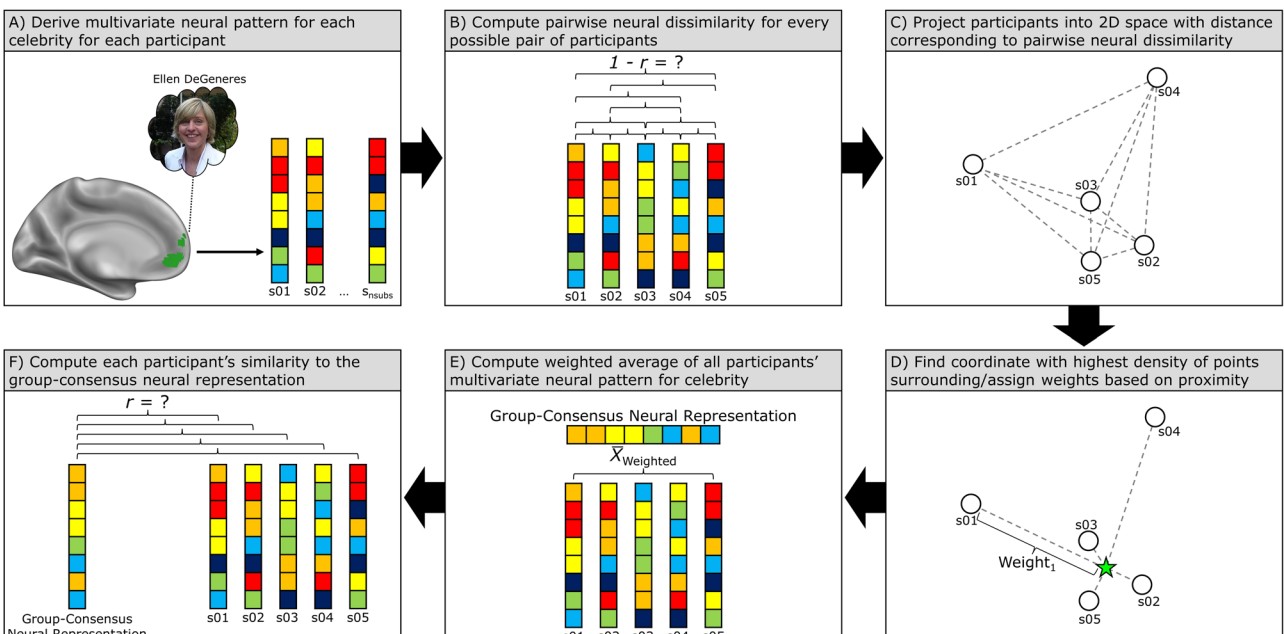

**Fig. 1 | Schematic depicting the group-consensus neural analyses undertaken in Study 1 A and Study 1B. A** During the fMRI task, participants reflected on the traits of a prominent celebrity (e.g., Ellen DeGeneres). Each participant's neural representation of a celebrity was defined as the vectorized parameter estimates across all voxels in a region of interest (e.g., the medial prefrontal cortex, shown in green below) while reflecting on that celebrity. **B** Next, the pairwise neural dissimilarity (i.e., correlation distance) was computed between every possible pair of participants' neural representations of a celebrity. **C** Participants were then projected into two-dimensional space using multidimensional scaling (MDS) with the distance between them corresponding to their pairwise neural dissimilarity. **D** A Gaussian kernel was then used on the MDS solution to identify the coordinate within the plane with the

greatest density of points surrounding it (denoted by the green star). Participants were then given a weight based on their proximity to this coordinate with greater weight being given to closer participants. **E** Next, using these weights, the group-consensus neural representation was computed as the weighted average of all participants' neural representations of a celebrity. **F** Finally, the Pearson correlation was computed between each participant's neural representation of a celebrity and the group-consensus neural representation of that celebrity. Surface rendering of medial prefrontal cortex region of interest made using Connectome Workbench (https://www.humanconnectome.org/software/connectome-workbench). Photo credit: Alan Light (retrieved from https://commons.wikimedia.org/wiki/File:Ellen_DeGeneres.jpg).

have idiosyncratic representations of celebrities compared not simply to other individuals but specifically to the group consensus. In addition to answering conceptually distinct questions, the Anna Karenina model and group-average model are not mutually inclusive statistically (i.e., if one is statistically significant, it need not be the case that the other is also statistically significant), as demonstrated through simulations with randomly generated data (see Supplementary Information: Supplementary Results pages 2–3, Supplementary Methods pages 22–23, and Supplementary Figs. 1–3).

Next, we tested if loneliness is associated with communication about well-known celebrities that deviates from the norm. Communication plays a key role in fostering social connection by creating a shared reality between people[17]. For example, when social ties gossip to create shared impressions of others[18] and when speakers tune their messages to fit audience members' perspectives[19], they end up feeling more connected to those with whom they have formed a common view. These observations, paired with prior work showing celebrities that generate common ground are disproportionately discussed in conversations[11], generated our next hypothesis: lonely individuals' communication about celebrities deviates from the zeitgeist. To test this, in Study 2, a new sample of participants described a well-known celebrity to a friend. To objectively measure whether lonelier participants communicated about celebrities in ways that deviated from the zeitgeist, we employed semantic similarity analysis. Mirroring our neural analyses, we tested whether lonelier participants were idiosyncratic in how they communicated their perceptions of celebrities compared to (A) other individuals and (B) the semantic group consensus. Overall, results supporting our hypotheses from this multi-method approach—ranging from neural to semantic similarity—would provide robust evidence that lonely individuals have idiosyncratic representations of well-known celebrities (one facet of contemporary cultural knowledge) that stray from the zeitgeist.

## Methods

### Study 1 A & 1B – participants

Study 1 A and 1B were approved by the Committee for the Protection of Human Subjects at Dartmouth College. All participants in Study 1 A and 1B reported normal neurologic history and normal or corrected-to-normal visual acuity, provided informed consent in accordance with the guidelines set by the Committee for the Protection of Human Subjects at Dartmouth college, and received compensation for their participation ($20.00/hour; introductory psychology students had the option to receive course credit instead). Study 1 A leveraged a dataset with previously published findings[20] (note that all previously reported analyses are orthogonal to those reported here). Fifty MRI-compatible participants (30 women, 20 men; aged 18–47 years, mean = 20.2 years) completed the study. Seven participants were excluded from Study 1 A due to movement in the scanner. Three more participants were excluded for whom there were no Revised UCLA Loneliness Scale[1] scores, leaving a sample of 40 participants for Study 1 A. Study 1B included an independent dataset of 48 MRI-compatible participants (32 women, 16 men; aged 18–30 years, mean = 21.0 years). Seven participants were excluded from Study 1B due to movement in the scanner. An additional participant was excluded due to a programming error during the task, leaving a sample of 40 participants for Study 1B.

### Study 1 A & 1B – procedure

While undergoing fMRI, participants completed a trait evaluation task for 16 targets: the self, 5 self-selected close others (e.g., friends, family members), 5 self-selected acquaintances (e.g., classmates, neighbors), and 5 well-known celebrities/public figures (Justin Bieber, Ellen DeGeneres, Kim Kardashian, Barack Obama, and Mark Zuckerberg) who were chosen due to their extremely high levels of fame and cultural significance as determined by Wikipedia search frequencies in prior work assessing shared cultural knowledge[3]. On each trial of the task, the name of a target was presented above a fixation crosshair and a trait adjective was presented below it. In Study 1 A, participants were instructed to rate how well the presented trait adjective described the target person on a scale from 1 (*not at all*) to 4 (*very*

*much*) using an MRI-compatible button box. In Study 1B, participants made a binary decision of 1 (*not applicable*) or 2 (*applicable*) as to how well the trait adjective described the target using an MRI-compatible button box. Fifty trials were completed per target across 10 functional runs in Study 1 A and across 4 functional runs in Study 1B. In Study 1B, the trait adjectives used in the task were chosen to reflect the Big Five Personality Dimensions[21] with 10 trait adjectives per dimension. In Study 1 A, the trait adjectives chosen were not restricted to these five dimensions. In both studies, an equal number of positively and negatively valenced trait adjectives were used. PsychoPy[22] was used for stimuli presentation and collection of timing data. An Epson (model ELP-7000) LCD projector displayed the stimuli on a screen at the head end of the scanner bore which participants viewed through a mirror mounted on the head coil.

Participants also rated each target outside the scanner on subjective closeness, similarity to self, and familiarity using a 0–100 scale ranging from *not at all* to *very much*. In addition, participants completed the Revised UCLA Loneliness scale[1] (Study 1 A: mean = 40.73, s.d. = 8.32, range = 25 to 57; Study 1B: mean = 42.43, s.d. = 10.09, range = 21–66). In Study 1 A, loneliness scores were not statistically significantly correlated with ratings of closeness (Ellen DeGeneres: $r_{(38)} = 0.02$, $p = 0.93$, 95% CI = −0.30 to 0.33; Kim Kardashian: $r_{(38)} = 0.24$, $p = 0.13$, 95% CI = −0.08 to 0.52; Barack Obama: $r_{(38)} = 0.15$, $p = 0.36$, 95% CI = −0.17 to 0.44; Justin Bieber: $r_{(38)} = 0.05$, $p = 0.76$, 95% CI = −0.27 to 0.36; Mark Zuckerberg: $r_{(38)} = 0.15$, $p = 0.37$, 95% CI = −0.17 to 0.44), similarity to self (Ellen DeGeneres: $r_{(38)} = 0.01$, $p = 0.97$, 95% CI = −0.31 to 0.32; Kim Kardashian: $r_{(38)} = 0.26$, $p = 0.11$, 95% CI = −0.06 to 0.53; Barack Obama: $r_{(38)} = 0.06$, $p = 0.69$, 95% CI = −0.25 to 0.37; Justin Bieber: $r_{(38)} = 0.19$, $p = 0.25$, 95% CI = −0.13 to 0.47; Mark Zuckerberg: $r_{(38)} = 0.07$, $p = 0.65$, 95% CI = −0.24 to 0.38), or familiarity (Ellen DeGeneres: $r_{(38)} = −0.03$, $p = 0.85$, 95% CI = −0.34 to 0.28; Kim Kardashian: $r_{(38)} = 0.05$, $p = 0.74$, 95% CI = −0.26 to 0.36; Barack Obama: $r_{(38)} = −0.01$, $p = 0.97$, 95% CI = −0.32 to 0.31; Justin Bieber: $r_{(38)} = 0.02$, $p = 0.91$, 95% CI = −0.30 to 0.33; Mark Zuckerberg: $r_{(38)} = 0.01$, $p = 0.95$, 95% CI = −0.30 to 0.32) for any of the target celebrities. In Study 1B, loneliness scores were not statistically significantly correlated with ratings of closeness (Ellen DeGeneres: $r_{(38)} = .09$, $p = 0.56$, 95% CI = −0.22 to 0.39; Kim Kardashian: $r_{(38)} = 0.05$, $p = 0.79$, 95% CI = −0.27 to 0.35; Justin Bieber: $r_{(37)} = 0.11$, $p = 0.51$, 95% CI = −0.21 to 0.41), similarity to self (Ellen DeGeneres: $r_{(38)} = −0.08$, $p = 0.63$, 95% CI = −0.38 to 0.24; Kim Kardashian: $r_{(38)} = −0.08$, $p = 0.64$, 95% CI = −0.38 to 0.24; Justin Bieber: $r_{(38)} = 0.04$, $p = 0.80$, 95% CI = −0.27 to 0.35), or familiarity (Ellen DeGeneres: $r_{(38)} = −0.02$, $p = 0.91$, 95% CI = −0.33 to 0.29; Kim Kardashian: $r_{(38)} = −0.07$, $p = 0.65$, 95% CI = −0.38 to 0.24; Justin Bieber: $r_{(38)} = 0.01$, $p = 0.96$, 95% CI = −0.30 to 0.32) for any of the target celebrities. In Study 1 A, these survey measures were completed directly after the scan whereas they were completed approximately 24 h before scanning in Study 1B. In both Study 1 A and Study 1B, several participants provided familiarity ratings of zero for one or more of the celebrities. However, all reported results remain consistent when data from these participants are excluded for these celebrities (see Supplementary Information: Supplementary Results pages 3–4).

### Study 1 A & 1B – fMRI image acquisition

Imaging data were acquired on a 3 T Siemens MAGNETOM Prisma Scanner (Siemens) with a 32-channel head coil. An anatomic (T1) image was acquired using a high-resolution 3-D MPRAGE sequence (TR = 2.3 s; TE = 2.3 ms; flip angle = 8°; 1 × 1 X 1 mm³ voxels). Functional images were collected using a T2*-weighted EPI sequence (TR = 1 s; TE = 30 ms; flip angle = 59°; echo spacing = 0.49; 2.5 × 2.5 × 2.5 mm resolution) with a simultaneous multi-slice of four and generalized auto-calibrating partial parallel acquisition of 1. In Study 1 A, ten functional runs of 250 axial images (52 slices, 130 mm coverage) were acquired for each participant. Sequence optimization was conducted using optseq2[23] and included approximately 30% jittered trials of fixation for obtaining a baseline estimation of neural activity. In Study 1B, four functional runs of either 510 (first 19 participants) or 590 (remaining 29 participants) axial images (52 slices, 130 mm coverage)

were acquired for each participant. Sequence optimization was conducted using easy-optimize-x (http://www.bobspunt.com/easy-optimize-x/) and included approximately 22% (first 19 participants) or 32% (remaining 29 participants) jittered fixation for obtaining a baseline estimation of neural activity.

## Study 1 A & 1B – fMRI preprocessing

Results included in this manuscript come from preprocessing performed using the default settings of fMRIPrep[24]. As recommended by Esteban and colleagues[24], for transparency and reproducibility we provide fMRIPrep's boilerplate text below unchanged with only minor edits for clarity (e.g., changing language to reflect steps were undertaken for all participants in the fMRI samples).

Results included in this manuscript come from preprocessing performed using fMRIPrep 21.0.2[24,25] (RRID:SCR_016216), which is based on Nipype 1.6.1[26,27] (RRID:SCR_002502).

*Preprocessing of $B_0$ Inhomogeneity Mappings.* $B_0$ nonuniformity maps (or *fieldmaps*) were estimated from the phase-drift map(s) measure with two consecutive GRE (gradient-recalled echo) acquisitions. The corresponding phase-map(s) were phase-unwrapped with prelude (FSL 6.0.5.1:57b01774).

*Anatomical Data Preprocessing.* T1-weighted (T1w) images were corrected for intensity non-uniformity (INU) with N4BiasFieldCorrection[28], distributed with ANTs 2.3.3[29] (RRID:SCR_004757), and used as T1w-references throughout the workflow. Each T1w-reference was then skull-stripped with a *Nipype* implementation of the antsBrainExtraction.sh workflow (from ANTs), using OASIS30ANTs as target template. Brain tissue segmentation of cerebrospinal fluid (CSF), white-matter (WM) and gray-matter (GM) was performed on each brain-extracted T1w image using fast[30] (FSL 6.0.5.1:57b01774, RRID:SCR_002823). Volume-based spatial normalization to one standard space (MNI152NLin2009cAsym) was performed through nonlinear registration with antsRegistration (ANTs 2.3.3), using brain-extracted versions of both T1w reference and the T1w template. The following template was selected for spatial normalization: *ICBM 152 Nonlinear Asymmetrical template version 2009c*[31] (RRID:SCR_008796; TemplateFlow ID: MNI152NLin2009cAsym).

*Functional Data Preprocessing.* For each of the 10 or 4 BOLD runs found per subject (across all tasks and sessions) for Study 1 A and Study 1B, respectively, the following preprocessing was performed. First, a reference volume and its skull-stripped version were generated using a custom methodology of *fMRIPrep*. Head-motion parameters with respect to the BOLD reference (transformation matrices, and six corresponding rotation and translation parameters) are estimated before any spatiotemporal filtering using mcflirt[32] (FSL 6.0.5.1:57b01774). The BOLD time-series (including slice-timing correction when applied) were resampled onto their original, native space by applying the transforms to correct for head-motion. These resampled BOLD time-series will be referred to as *preprocessed BOLD in original space*, or just *preprocessed BOLD*. The BOLD reference was then co-registered to the T1w reference using mri_coreg (FreeSurfer) followed by flirt[33] (FSL 6.0.5.1:57b01774) with the boundary-based registration[34] cost-function. Co-registration was configured with six degrees of freedom. Several confounding time-series were calculated based on the *preprocessed BOLD*: framewise displacement (FD), DVARS and three region-wise global signals. FD was computed using two formulations following Power[35] (absolute sum of relative motions) and Jenkinson[32] (relative root mean square displacement between affines). FD and DVARS are calculated for each functional run, both using their implementations in *Nipype* (following the definitions by Power et al.[35]). The three global signals are extracted within the CSF, the WM, and the whole-brain masks. Additionally, a set of physiological regressors were extracted to allow for component-based noise correction[36] (*CompCor*). Principal components are estimated after high-pass filtering the *preprocessed BOLD* time-series (using a discrete cosine filter with 128 s cut-off) for the two *CompCor* variants: temporal (tCompCor) and anatomical (aCompCor). tCompCor components are then calculated from the top 2% variable voxels within the brain mask. For aCompCor, three probabilistic masks (CSF, WM and combined CSF +

WM) are generated in anatomical space. The implementation differs from that of Behzadi et al.[36] in that instead of eroding the masks by 2 pixels on BOLD space, the aCompCor masks are subtracted a mask of pixels that likely contain a volume fraction of GM. This mask is obtained by thresholding the corresponding partial volume map at 0.05, and it ensures components are not extracted from voxels containing a minimal fraction of GM. Finally, these masks are resampled into BOLD space and binarized by thresholding at 0.99 (as in the original implementation). Components are also calculated separately within the WM and CSF masks. For each CompCor decomposition, the $k$ components with the largest singular values are retained, such that the retained components' time series are sufficient to explain 50 percent of variance across the nuisance mask (CSF, WM, combined, or temporal). The remaining components are dropped from consideration. The head-motion estimates calculated in the correction step were also placed within the corresponding confounds file. The confound time series derived from head motion estimates and global signals were expanded with the inclusion of temporal derivatives and quadratic terms for each[37]. Frames that exceeded a threshold of 0.5 mm FD or 1.5 standardised DVARS were annotated as motion outliers. The BOLD time-series were resampled into standard space, generating a *preprocessed BOLD run in MNI152NLin2009cAsym space*. First, a reference volume and its skull-stripped version were generated using a custom methodology of *fMRIPrep*. All resamplings can be performed with *a single interpolation step* by composing all the pertinent transformations (i.e. head-motion transform matrices, susceptibility distortion correction when available, and co-registrations to anatomical and output spaces). Gridded (volumetric) resamplings were performed using antsApplyTransforms (ANTs), configured with Lanczos interpolation to minimize the smoothing effects of other kernels[38]. Non-gridded (surface) resamplings were performed using mri_vol2surf (FreeSurfer).

## Study 1 A & 1B – fMRI response estimation

GLMs were conducted using nltools (nltools.org) with each of the 16 individual identities (i.e., the self, 5 close others, 5 acquaintances, and 5 celebrities) defined as a separate condition. GLMs incorporated nuisance regressors (the 6 standard motion parameters and their derivatives, the signal extracted from white matter regions, the signal extracted from cerebrospinal fluid regions, and a high-pass filter (128 s)) and were convolved with a canonical hemodynamic response function (HRF) to compute parameter estimates ($\beta$) and contrast images (containing weighted parameter estimates) at each voxel. Two GLMs were conducted each using only half the data (i.e., only odd or even runs) for the purpose of reliability-based voxel selection[39]. A third GLM included all trials and was used for all subsequent analyses.

## Study 1 A – reliability-based voxel selection

Unless otherwise specified, all analyses described below were conducted using the PyMVPA toolbox[40] within JupyterLab[41]. Reliability-based voxel selection[39] (RBVS) was implemented to identify regions of the brain in which identity-specific information is reliably represented. The RBVS procedure was conducted only on the data from Study 1 A. In the first step of RBVS, the vector of parameter estimates for the 16 conditions included in the study in one half of the data was correlated with the corresponding vector in the other half of the data at every voxel, yielding a whole-brain map of voxel-wise reliability. Individual participant maps of voxel-wise reliability were averaged together into a single group map of voxel-wise reliability. The second step proceeds by examining the average condition multivoxel pattern reliability across increasingly stringent thresholds of voxel-wise reliability. That is, for each condition, the multivoxel pattern of neural activity across all voxels exceeding a certain voxel-wise reliability threshold in one half of the data is correlated with the corresponding multivoxel pattern of neural activity in the other half of the data, and then the condition multivoxel pattern reliability is averaged across all conditions. Finally, the average condition multivoxel pattern reliability values across all participants were averaged together into a single group estimate of average condition

multivoxel pattern reliability corresponding to each voxel-wise reliability threshold examined. Because previous work has highlighted the medial prefrontal cortex (MPFC) and the precuneus (PC)/posterior cingulate cortex (PCC) as central to the neural representations of people[20,42,43], the second step of RBVS was modified to focus only on these two regions of interest. That is, rather than computing condition multivoxel pattern reliability across all voxels exceeding a given threshold, it was computed only for those voxels exceeding a given threshold that formed a contiguous cluster that included the peak voxel (i.e., with respect to voxel-wise reliability) in either the MPFC or PC/PCC.

In RBVS, voxel-wise reliability is plotted against average condition multivoxel pattern reliability and compared alongside brain maps to identify a threshold that balances each type of reliability along with reasonable coverage of the brain regions of interest. In the current study, average condition multivoxel pattern reliability was plotted for each region of interest (ROI) separately against voxel-wise reliability thresholds ranging from $r = 0.160$ to $r = 0.420$. Because the PC/PCC and much of the visual cortex formed a single, large contiguous cluster at lower thresholds, .160 was chosen as the lower end of the range examined in order to focus the RBVS procedure only on the PC/PCC and not the visual cortex. The upper end of the range examined was defined as the threshold above which the size of either ROI dropped below 33 voxels (the size of a 3-voxel radius spherical searchlight). As can be seen in Supplementary Fig. 4, the pattern of change in multivoxel pattern reliability across different voxel-wise reliability thresholds was not consistent across the two ROIs. Therefore, to optimize multivoxel pattern reliability in a way that took both ROIs into account, we z scored multivoxel pattern reliability in each ROI across all voxel-wise reliability thresholds examined. Then, we averaged the z scored reliability values and identified the voxel-wise reliability threshold with the maximum average z scored reliability. This process identified a voxel-wise reliability threshold of 0.344 as the one that best balanced average condition multivoxel pattern reliability across the two ROIs, yielding a 208-voxel ROI in the MPFC and a 525-voxel ROI in the PC/PCC (Supplementary Fig. 4). All subsequent analyses in Study 1 A were focused on these two ROIs. Because statistically significant results were found only in the MPFC in Study 1 A, replication analyses in Study 1B were focused only on the 208-voxel MPFC ROI identified using the RBVS procedure on the data from Study 1 A.

### Study 1 A & 1B – pairwise multivariate neural similarity

Pairwise similarity in neural representations of celebrities was calculated by vectorizing the parameter estimates in an ROI for a given celebrity and correlating these vectors across every possible pair of participants. This approach has been used previously in round-robin fMRI studies in which the targets in a trait evaluation task are common across all participants[13,16]. Because we hypothesized that lonelier individuals would be especially idiosyncratic in their neural representations of well-known public figures, we modeled loneliness as each pair's mean loneliness score, which is an example of an Anna Karenina model[12]. Rather than predicting that neural similarity is associated with similarity on the behavioral measure of interest regardless of which end of the scale a pair trends toward (as is the case when modeling pairs' scores in terms of dis/similarity), Anna Karenina models instead test whether one or the other end of the scale tends to be higher in neural similarity relative to the other. In this case, a negative association between pairwise neural similarity and mean loneliness scores would indicate that lonelier pairs were especially dissimilar from one another, as well as relative to less lonely participants, in their neural representations of a celebrity.

To account for the non-independence of the data (i.e., each participant is represented in multiple pairs) as well as the nested nature of the data (i.e., there were multiple observations per pair, one per celebrity), linear mixed-effects modeling was implemented in the R statistical language using the Lme4 package[44] with a random intercept for the first participant in a pair, the second participant in a pair, and the celebrity. The lmerTest package[45] was used to calculate Satterthwaite approximated degrees of freedom and corresponding p values for all linear mixed-effects models reported. Confidence

intervals were estimated using the confint function of the Lme4 package (1000 simulations). Consistent with previous work examining the association between pairwise neural similarity and loneliness[7], the observations for each pairwise analysis were doubled to allow each participant to be modeled as both the first and second participant in a pair for each observation[46]. Doubling the data in this manner allows for fully crossed random effects and accounts for the symmetric nature of the neural similarity measure. In other words, doubling the data allows for a random intercept for each participant (as both participant 1 and participant 2) with the appropriate number of observations (i.e., N – 1, the number of pairs each participant contributes to), whereas otherwise the participant 1 and participant 2 random effects would not be symmetric. The resulting redundancy in the data was then accounted for by halving the estimated degrees of freedom and adjusting the p value accordingly. For interpretability of beta values and comparison across models, all variables were z scored prior to conducting each linear mixed-effects analysis. Data distributions were visually inspected for normality (for histograms of outcome variables, see Supplementary Fig. 5 and Supplementary Fig. 7), but this was not formally tested. To test moderation by level of consensus, we first ensured that there was in fact a statistically significant difference across celebrities in their distribution of pairwise neural similarity by conducting t tests for every possible pair of celebrities examined. Again, to account for the non-independence of the data these t tests were conducted using linear mixed-effects modeling following the same procedure described above.

### Study 1 A & 1B – similarity to group-consensus neural representations

Next, we tested the question of whether lonelier individuals were not only especially different from one another but from a group-consensus neural representation of each celebrity. To define the group consensus neural representation of each celebrity, multidimensional scaling (MDS) was implemented using scikit-learn's[47] manifold function with four initializations and a maximum of 3000 iterations per run to project participants into coordinates in two-dimensional space with the distance between these coordinates determined based on the correlation distance between participants' neural representations of a given celebrity. Next, a Gaussian kernel was used to identify the coordinate within this plane with the highest density of surrounding points (the radius of the kernel was defined as the distance between the maximum and minimum value along either the x or y axis, whichever was smaller). Once the coordinate of the densest point in the plane was identified, each participant was weighted according to their proximity to this point. Specifically, the Euclidean distance was calculated between each participant's coordinate and the coordinate of the densest point in the plane. Then, to recode these values as proximity rather than distance, each participant's Euclidean distance value was subtracted from the highest Euclidean distance value, thus giving the furthest participant from this point a weight of zero. The group consensus neural representation of a celebrity was then calculated as the weighted mean across participants of their z-scored neural pattern of activity for that celebrity with the weights being defined as just described above. Finally, each participant's neural representation of a celebrity was correlated with the group-consensus neural representation of that celebrity. To investigate the association between loneliness scores and participants' similarity to the group consensus, linear mixed-effects modeling was implemented with a random intercept for participant (because models were overfitted when also including a random intercept for celebrity, only random intercepts for participants were included in this case). Figure 1 depicts a schematic of the analysis described in this paragraph. Data distributions were visually inspected for normality and were found in this case to exhibit some negative skew (for histograms of outcome variables, see Supplementary Fig. 6 and Supplementary Fig. 8), but normality was not formally tested.

We note here that one downside to projecting participants into two-dimensional space using MDS is that some information is lost in the process. Despite this drawback, we implemented the approach described in the preceding paragraph for two main reasons. First, compared to other options,

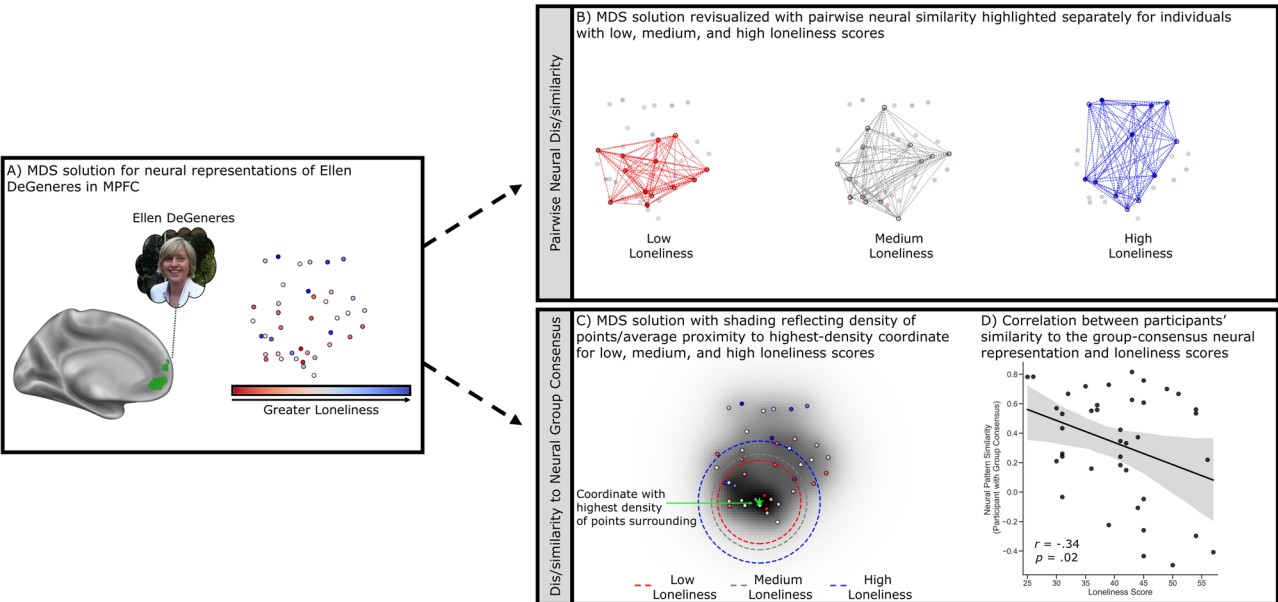

**Fig. 2 | The results of Study 1 A for one target celebrity, Ellen DeGeneres ($N = 40$ participants).** **A** Left medial surface rendering of the region of interest in the medial prefrontal cortex (MPFC, shown in green) and participants projected into two-dimensional space using multidimensional scaling (MDS) with greater proximity between points reflecting more similar neural representations of Ellen DeGeneres in the MPFC. Points are colored according to loneliness scores with darker blue reflecting higher scores (i.e., more loneliness) and darker red reflecting lower scores (i.e., less loneliness). **B** A trinary split on loneliness scores was used to revisualize the MDS solution with pairwise neural similarity highlighted separately for individuals with low (red dotted lines), medium (gray dotted lines), and high (blue dotted lines) loneliness scores. Of note, the High Loneliness participants (blue) are more distant from one another in the MDS plot, indicating less overall pairwise neural similarity relative to the Low Loneliness participants (red) who are more densely packed together. **C** The MDS solution overlaid on the results of the Gaussian kernel used to identify the densest coordinate in the plane. Areas with a greater density of points

surrounding are shown with darker shading and the coordinate with the highest density of points surrounding is denoted with a green star. A trinary split on loneliness scores was used to show the average distance from the densest coordinate in the plane for individuals with low (red circle), medium (gray circle), and high (blue circle) loneliness scores. **D** A scatterplot showing the negative relationship between loneliness scores (x-axis) and the similarity between participants' neural representations of Ellen DeGeneres and the group-consensus neural representation of her (y-axis), defined as the weighted average of participants' neural representations of her with greater weight being given to participants closer to the densest point in the MDS plot (i.e., the green star). Shaded 95% confidence interval for regression estimate derived through bootstrapping. Surface rendering made using Connectome Workbench (https://www.humanconnectome.org/software/connectome-workbench). Photo credit: Alan Light (retrieved from https://commons.wikimedia.org/wiki/File: Ellen_DeGeneres.jpg).

we think it better captures the idea of the zeitgeist because participants are weighted more heavily the closer they are to a single point of greatest convergence across all participants. Second, projecting participants into two-dimensional space greatly facilitates plotting the distance between all participants in the sample, allowing one to visually inspect whether there is in fact a single point at which many participants appear to converge (see Fig. 2 for an example). Nonetheless, to ensure that the results of this analysis did not hinge on this decision, we additionally report all results using two alternative approaches: the unweighted average of all participants' neural representations of a celebrity, and the distance-weighted average of all participants' neural representations of a celebrity which does not require MDS (i.e., participants are weighted according to their average correlation distance from all other participants).

### Study 1 A & 1B – ruling out the potential role of emotional closeness to celebrities

Previous research demonstrates that neural representations of people in midline cortical structures are organized according to their psychological closeness[20,42], and some evidence suggests that lonelier individuals are more likely to seek the experience of social connection from watching TV and may therefore be more likely to form parasocial attachments to media figures[48,49]. Therefore, we also examined ratings of closeness with celebrities to assess whether any associations between loneliness and more idiosyncratic neural representations of celebrities were attributable to greater emotional attachment to celebrities. For the pairwise analyses, we tested for an Anna Karenina effect (i.e., closeness was modeled as each pair's mean rating) as was done for loneliness.

### Study 1B – controlling for participant behavior during the fMRI trait-rating task

Participant responses during the fMRI trait-rating task were not recorded in Study 1 A. However, in Study 1B we examined and controlled for participants' behavior during the fMRI task (i.e., their trait judgments, the speed of their response times, and the number of trials they missed). Because for each trait rating participants made a binary decision (i.e., the trait displayed is "applicable" or "not applicable" to the target person displayed), we calculated the Jaccard similarity between every possible pair of participants' trait-task responses for a given celebrity after dropping traits with missing responses for either participant. To examine individual participants' similarity to the group-consensus trait perceptions of celebrities we calculated the unweighted average response across all participants for each of the 50 traits used in the task. For each participant, we then calculated the point biserial correlation between their binary responses to the trait-rating task and the group average responses after dropping any missing trials for that participant.

Pairwise dissimilarity in number of missing responses and response time was modeled as the absolute difference in these values for every possible pair of participants. Pairwise dissimilarity was converted to pairwise similarity by taking the inverse of the *z-scored* dissimilarity values. Response times were excluded from one participant for whom there was a software error in the recording of these values.

For the control analyses described in this section, Bayes Factors (BF) were calculated for analyses that were not statistically significant with null hypothesis significance testing to assess the strength of the evidence in favor of the null hypothesis. Bayes Factors were calculated using the BayesFactor package in the R statistical language with priors defined by default settings.

## Study 1 A & 1B – statistical significance testing

Based on work demonstrating that lonelier individuals have more idiosyncratic neural responses to media[7] and that those with fewer close connections have more idiosyncratic neural representations of their peers[13], our a priori hypothesis was that lonelier individuals would have more idiosyncratic neural representations of celebrities. Therefore, we report one-tailed *p* values for all analyses testing this hypothesis. (We further note that the main analyses in Study 1 A that are replicated in Study 1B are statistically significant in both studies even with two-sided tests.) For all other tests (e.g., control analyses, and exploratory or post hoc tests for moderation), we report two-tailed *p* values.

## Study 2 – participants

Study 2 was approved by the Committee for the Protection of Human Subjects at Dartmouth College. One-thousand-thirty-five participants were recruited using Amazon's Mechanical Turk (MTurk). All participants in Study 2 provided informed consent in accordance with the guidelines set by the Committee for the Protection of Human Subjects at Dartmouth College and received compensation for their participation ($4.00 for an estimated 10-min study). The online survey consisted of several checks designed to exclude data from bots and inattentive participants, including the use of Captcha at the beginning of the survey and ensuring they provided the same age at two different points in the survey. Because the study required that participants write free-response paragraphs, they were asked to type "I will answer open-ended questions" into a free-response text box, and those who failed to do so were excluded. In addition, two questions were repeated with the valence of the wording flipped (e.g., "*I dislike [name of celebrity]*" rather than "*I like [name of celebrity]*"), and we ensured that participants were paying sufficiently close attention to adjust their answers accordingly. We also included two questions asking participants to move a slider (the format in which they provided ratings of the celebrities) to a particular number (e.g., "Please move the slider to the number ten") to ensure participants were carefully reading each question before providing their responses. Free-response paragraphs were checked to make sure participants wrote about the correct celebrity. Data from 87 participants was excluded based on the criteria described above. Data from an additional 13 participants was excluded for whom there was no variance in their responses for the celebrity ratings. Variance in participants' celebrity ratings was *z-scored* for the remaining participants and data from an additional 12 participants was dropped who were two or more standard deviations below the mean variance. Following these exclusions, we analyzed a final sample of 923 participants (mean age = 40.0 years, range: 18–77 years; 45.0% women, 53.9% men, 1.0% non-binary; 6.5% Asian, 10.1% Black, 4.8% Latinx, 2.9% Multiracial, 0.9% Native American, 74.9% White).

## Study 2 – procedure

Participants completed the Revised UCLA Loneliness Scale[1] and a series of questions about a prominent celebrity. The order in which participants completed the loneliness measure and the celebrity portion of the survey was randomized. The celebrities selected as targets in Study 2 were drawn from the same pool of 60 well-known public figures (built based on Wikipedia search frequencies)[3] used to select the target celebrities in Study 1. Because the results of the pairwise analyses in Study 1 A suggested that the effect was driven primarily by pop culture celebrities rather than prominent figures in politics or business, we included the three pop culture celebrities from Study 1 (Justin Bieber, Ellen DeGeneres, and Kim Kardashian) along with seven others: Cameron Diaz, Harrison Ford, Megan Fox, Michael Jordan, Keanu Reeves, Will Smith, and Justin Timberlake. In the celebrity portion of the survey, participants were first asked to indicate which of the 10 celebrities they had previously heard of. The remainder of the celebrity portion of the survey then focused on one celebrity who was randomly selected for each participant from only those celebrities with which they were familiar. Participants wrote a paragraph describing their celebrity after reading the following prompt:

"*Imagine that a friend of yours has never heard of [name of celebrity]. In the box below, please write a paragraph describing to your friend in as much detail as possible who [name of celebrity] is. Try not to describe [name of celebrity] solely in terms of their career or accomplishments. Instead, try to communicate your unique perspective on who they are as a person (e.g., how you feel about them, your perception of their characteristics, etc.).*" Participants also rated their agreement with statements reflecting psychological closeness to the celebrity on a 0 (*Disagree strongly*) to 100 (*Agree strongly*) scale, including their liking of the celebrity, how similar they perceived the celebrity to be to themselves, how close they felt to the celebrity, and how much they felt they knew about the celebrity. Participants also provided ten trait ratings (trait dimensions: openness, conscientiousness, extraversion, agreeableness, neuroticism, warmth, competence, trustworthiness, dominance, and intelligence) for the celebrity by indicating their agreement on a 0 (*Disagree strongly*) to 100 (*Agree strongly*) scale with the following statement: "*I see [name of celebrity] as [trait].*" Lastly, participants rated their agreement with the following two statements on the same 0 to 100 scale: "*My perception of [name of celebrity] is similar to those of the people around me*" and "*My perception of [name of celebrity] is accurate.*" The order in which participants completed the ratings and wrote the paragraph was randomized.

## Study 2 – pairwise semantic similarity

Pairwise semantic similarity was determined by first embedding participants' paragraphs describing their celebrity into a common 512-dimensional space using Google's Universal Sentence Encoder (USE)[50], a natural language processing tool that reflects semantic content. We then calculated the cosine similarity between each pair's USE-derived vectors for every possible pair matching in the celebrity they described. In previous research, USE has been demonstrated to be an effective tool for capturing similar interpretations of narratives across participants based on their recalled descriptions[51]. It has also been used to determine which events in a narrative are most central to the story based on their semantic similarity to the other events in the story, which in turn predicted which events were better remembered[52]. Consistent with these studies, we interpret greater semantic similarity in pairs' descriptions of a celebrity as reflecting greater overlap in their subjective understanding of who that person is.

To test whether lonelier individuals were more idiosyncratic in the language they used to communicate their impressions of celebrities, we followed the same basic analytic procedures as in Study 1 using the same statistical software. We modeled loneliness as each pair's mean loneliness score. Linear mixed-effects modeling was again implemented to account for the non-independence inherent to pairwise analyses as well as the nested nature of the data (pairs of participants were nested within the celebrity they had both rated/written about) with a random intercept for the first participant in a pair, the second participant in a pair, and the celebrity. The observations for each pairwise analysis were doubled to allow each participant to be modeled as both the first and second participant in a pair for each observation[7,46]. The resulting redundancy in the data was then accounted for by halving the estimated degrees of freedom and adjusting the *p*-value accordingly. For interpretability of beta values and comparison across models, all variables were *z-scored* prior to conducting each linear mixed-effects analysis. Data distributions were visually inspected for normality (for histograms of outcome variables, see Supplementary Fig. 9), but this was not formally tested.

To test moderation by level of consensus, we first performed a median split on the ten celebrities based on their mean pairwise semantic similarity. We also conducted *t*-tests to examine the level of variability in distributions of pairwise semantic similarity values across the ten target celebrities. Again, to account for the non-independence of the data these *t*-tests were conducted using linear mixed-effects modeling following the same procedure described above.

In the case of significant results supporting the hypothesis that lonelier pairs are especially dissimilar in their communication about prominent

celebrities we conducted additional control analyses to assess whether loneliness was predictive of pairwise semantic similarity above and beyond other variables that might also account for shared perceptions of celebrities such as demographic factors. Similarity in categorical demographic variables (i.e., gender and race) was modeled such that a pair was coded as 1.0 if they matched in their self-reported gender or race and 0.0 if they did not. Continuous single-value control variables (i.e., age, word count, and ratings of liking, similarity, closeness, and familiarity) were modeled as the absolute difference in a pair's self-reported age, the word count of their paragraph/s, or in their ratings of psychological closeness to the celebrity. Dissimilarity in perceptions of a celebrity's traits was modeled as the Euclidean distance between a pair's ten trait ratings. To convert pairwise dissimilarity values to similarity values, the signs of pairwise dissimilarity values were simply flipped following $z$ scoring. The results of control analyses are reported in the Supplementary Information (Supplementary Results pages 4–6 and Supplementary Tables 2–4).

### Study 2 – similarity to the group-consensus semantic representations of celebrities

As was done in Study 1, we additionally tested the question of whether lonelier individuals were not only especially different from other individuals in terms of semantic similarity but from the group-consensus semantic representation of the celebrity they wrote about. The group consensus semantic representation of each celebrity was defined following the same steps used to define the group-consensus neural representations of celebrities in Study 1. MDS was implemented to project participants into coordinates in two-dimensional space with the distance between these coordinates based on the cosine distance between participants' USE-derived vectors. Next, a Gaussian kernel was used to identify the coordinate within this plane with the highest density of surrounding points (the radius of the kernel was defined as the distance between the maximum and minimum value along either the x or y axis, whichever was smaller). Once the coordinate of the densest point in the plane was identified, each participant was weighted according to their proximity to this point. Specifically, the Euclidean distance was calculated between each participant's coordinate and the coordinate of the densest point in the plane. Then, to recode these values as proximity rather than distance, each participant's Euclidean distance value was subtracted from the highest Euclidean distance value, thus giving the furthest participant from this point a weight of zero. The group-consensus semantic representation of a celebrity was then calculated as the weighted mean across participants of their 512 semantic features provided by USE with the weights being defined as just described above. Finally, the cosine similarity was calculated between every participant's USE-derived vector and the group-consensus semantic representation of the celebrity they wrote about. To investigate the association between loneliness scores and participants' similarity to the group consensus, linear mixed-effects modeling was implemented with a random intercept for celebrity. Data distributions were visually inspected for normality and were found in this case to exhibit some negative skew (for histograms of outcome variables, see Supplementary Fig. 10), but normality was not formally tested. As in Study 1, we additionally report results using two alternative approaches for calculating the group-consensus semantic representation of a celebrity: the unweighted average and the distance-weighted average (distance in this case being cosine distance).

### Study 2 – statistical significance testing

Because we were testing for the same direction in the pattern of results as was observed in Study 1 (i.e., more idiosyncratic representations of celebrities for lonelier individuals, and moderation by consensus such that higher consensus is associated with greater idiosyncrasy for lonelier individuals), we report one-tailed $p$ values for all analyses testing these hypotheses. For all other tests (e.g., control analyses), we report two-tailed $p$ values.

### Preregistration

The studies and associated analyses reported herein were not preregistered.

## Results

### Study 1 A – regions of interest

In Study 1 A, we tested the a priori hypothesis that lonelier individuals are more idiosyncratic in their neural representations of celebrities, both compared to other individuals and to group-consensus neural representations. We also explored the possibility that this association is moderated by the degree of consensus surrounding the group's neural representations of a given celebrity. As noted in the introduction, the consensus surrounding how these celebrities are perceived is part of the reason why straying from these commonly held views carries implications for one's general sense of belonging. The less you see the world in ways that everyone else agrees with, the lonelier you may feel. Thus, we expected that the association between loneliness and more idiosyncratic neural representations of celebrities would be especially strong for those celebrities who were more similarly represented across the sample.

Our sample consisted of 40 participants who completed a trait evaluation task for the same set of five celebrities while undergoing fMRI: Ellen DeGeneres, Kim Kardashian, Justin Bieber, Barack Obama, and Mark Zuckerberg. These celebrities were selected because of their extremely high levels of popularity as determined by Wikipedia search frequencies in prior work assessing shared cultural knowledge[3]. In the task, participants saw the name of a celebrity above a trait adjective and made a judgment as to how well that word described that person for 50 different trait adjectives. Reliability-based voxel selection[39] was implemented to identify regions of the brain in which information about people is reliably represented (see methods section and Supplementary Fig. 4 for details). This process yielded two regions of interest (ROI), a 208-voxel ROI in the medial prefrontal cortex (MPFC) and a 525-voxel ROI in the precuneus (PC)/posterior cingulate cortex (PCC), consistent with prior research implicating these regions in representing person knowledge[20,42,43].

### Study 1 A – pairwise similarity in neural representations of celebrities

We first examined the Anna Karenina model, which tested whether neural representations of celebrities were especially dissimilar for lonelier participants relative to both other lonely participants and less lonely participants. Neural representational similarity was calculated by vectorizing the pattern of neural activity associated with a celebrity in each ROI for each participant and then computing the Pearson correlation between these vectors for every possible pair of participants (i.e., pairwise neural similarity). Loneliness was modeled as each pair's mean loneliness score. Consistent with our hypotheses, the results of a linear mixed-effects analysis indicated that greater mean loneliness scores were associated with decreased pairwise neural similarity in the MPFC ($\beta = -0.15$, SE = 0.05, $t_{(39)} = 2.95$, $p = 0.003$, 95% CI = $-0.24$ to $-0.04$). This was not the case, however, in the PC/PCC ($\beta = 0.06$, SE = 0.07, $t_{(38)} = 0.91$, $p = 0.82$, 95% CI = $-0.07$ to 0.20). So far, results indicate lonelier participants' neural representations of celebrities in the MPFC are particularly idiosyncratic. The top portion of Fig. 2 depicts this pattern of results for one of the target celebrities, Ellen DeGeneres.

Next, we explored whether the effect observed in the MPFC was moderated by the degree of neural consensus (i.e., average pairwise neural similarity) associated with each celebrity. The first step in answering this question was assessing whether neural representations of one or more celebrities elicited particularly strong neural consensus across all pairs. To this end, we examined the distribution of pairwise neural similarity for each celebrity. There was statistically significantly greater similarity across pairs in their neural representations of Justin Bieber relative to the other four celebrities (Justin Bieber vs. Ellen DeGeneres: $\beta = 0.04$, SE = 0.01, $t_{(1520)} = 4.90$, $p < 0.001$, 95% CI = 0.03 to 0.06; Justin Bieber vs. Kim Kardashian: $\beta = 0.03$, SE = 0.01, $t_{(1520)} = 3.77$, $p < 0.001$, 95% CI = 0.02 to 0.05; Justin Bieber vs. Barack Obama: $\beta = 0.03$, SE = 0.01, $t_{(1520)} = 3.59$, $p < 0.001$, 95% CI = 0.02 to 0.05; Justin Bieber vs. Mark Zuckerberg: $\beta = 0.03$, SE = 0.01, $t_{(1520)} = 3.26$, $p = 0.001$, 95% CI = 0.01 to 0.04), indicating particularly strong neural consensus for this celebrity. There was no significant difference in the distribution of pairwise neural similarity for the other four

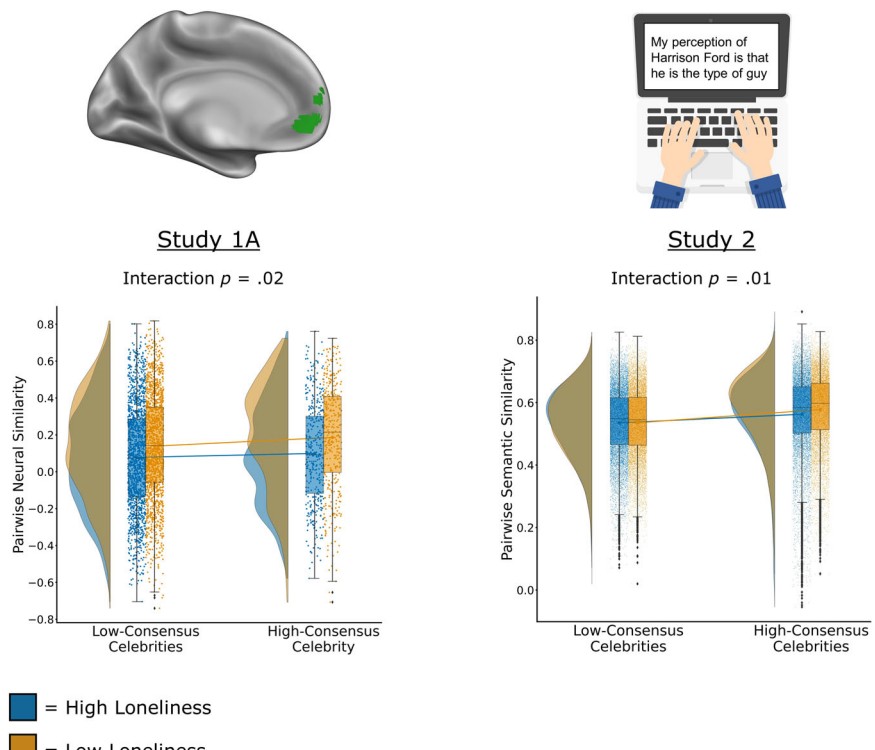

**Fig. 3 | Neural and semantic evidence that level of consensus moderates the association between loneliness and more idiosyncratic representations of celebrities.** Across two different measures of pairwise representational similarity – pairwise neural similarity in the medial prefrontal cortex (shown in green in top left) in Study 1 A and pairwise semantic similarity in Study 2 – we found that the association between mean loneliness scores and pairwise similarity was moderated by the level of consensus. Specifically, there was a stronger negative association between mean loneliness scores and pairwise similarity for celebrities for which there was a stronger consensus in neural patterns of activity when reflecting on that celebrity (Study 1 A) or a stronger consensus in terms of the language used when communicating one's perception of that celebrity (Study 2). In the plots below, a median split on mean loneliness scores is used to visualize the interactions. Boxplots display medians/quartiles; error bars extend 1.5*inter-quartile range beyond upper/lower quartiles. Study 1 A: $N = 3900$ observations (780 unique pairs derived from 40 participants X neural representations of 5 celebrities). Study 2: $N = 42,638$ observations/unique pairs (derived from 923 participants; 1 observation per unique pair as participants wrote a description of only one celebrity). Surface rendering made using Connectome Workbench (https://www.humanconnectome.org/software/connectome-workbench). Note that the text displayed on the laptop screen is not taken from an actual participant response but rather was invented by the authors for illustrative purposes. Clipart retrieved from https://commons.wikimedia.org/wiki/File:Hands_typing_on_white_laptop_scene.svg.

celebrities (Kim Kardashian vs. Ellen DeGeneres: $\beta = 0.01$, SE = 0.01, $t_{(1520)} = 1.22$, $p = 0.22$, 95% CI = $-0.01$ to 0.03; Barack Obama vs. Ellen DeGeneres: $\beta = 0.01$, SE = 0.01, $t_{(1520)} = 1.22$, $p = 0.22$, 95% CI = $-0.01$ to 0.03; Mark Zuckerberg vs. Ellen DeGeneres: $\beta = 0.02$, SE = 0.01, $t_{(1520)} = 1.74$, $p = 0.08$, 95% CI = $-0.003$ to 0.03; Barack Obama vs. Kim Kardashian: $\beta = 0.0005$, SE = 0.01, $t_{(1520)} = 0.05$, $p = .96$, 95% CI = $-0.02$ to 0.02; Mark Zuckerberg vs. Kim Kardashian: $\beta = 0.005$, SE = 0.01, $t_{(1520)} = 0.53$, $p = 0.59$, 95% CI = $-0.01$ to 0.02; Mark Zuckerberg vs. Barack Obama: $\beta = 0.004$, SE = 0.01, $t_{(1520)} = 0.47$, $p = 0.64$, 95% CI = $-0.01$ to 0.02). The second step in answering our question was to determine if lonelier participants showed idiosyncratic neural patterns for the high-consensus celebrity (Justin Bieber) to a greater degree than for low-consensus celebrities (Ellen DeGeneres, Kim Kardashian, Barack Obama, Mark Zuckerberg). A linear mixed-effects model was run to test whether the relationship between loneliness and pairwise neural similarity was moderated by neural consensus, with Justin Bieber categorized as a high-consensus celebrity (coded as 1.0) and the other four celebrities categorized as low-consensus celebrities (coded as $-1.0$). As can be seen in Fig. 3, results showed that the association between mean loneliness scores and pairwise neural similarity was indeed moderated by level of consensus as indicated by a significant interaction between mean loneliness scores and the contrast of high vs. low consensus: $\beta = -0.03$, SE = 0.01, $t_{(3858)} = 2.25$, $p = 0.02$, 95% CI = $-0.05$ to $-0.003$. That is, consistent with our prediction, the negative association between mean loneliness and pairwise neural similarity was stronger for the

high-consensus celebrity relative to the low-consensus celebrities. However, we additionally note that when level of consensus was modeled continuously (i.e., as the mean pairwise neural similarity for each celebrity), the interaction between mean loneliness scores and level of consensus was not statistically significant: $\beta = -0.01$, SE = 0.01, $t_{(3859)} = 1.26$, $p = 0.0.21$, 95% CI = $-0.03$ to 0.007. This is likely because there was limited variability in consensus across targets (i.e., four of the five celebrities were virtually identical in level of consensus as discussed above). There was no statistically significant interaction between mean loneliness and level of consensus in the PC/PCC, whether modeled as the contrast of high vs. low consensus ($\beta = 0.01$, SE = 0.01, $t_{(3858)} = 0.56$, $p = 0.58$, 95% CI = $-0.01$ to 0.02) or modeled continuously ($\beta = -0.0003$, SE = 0.01, $t_{(3859)} = 0.03$, $p = 0.97$, 95% CI = $-0.02$ to 0.02).

Examining the association between mean loneliness scores and pairwise neural similarity in the MPFC for each celebrity individually revealed a qualitative difference between those celebrities exhibiting a strong negative association and those exhibiting a relatively weaker one. Lonelier pairs of participants were especially dissimilar in their neural representations of pop culture celebrities (Justin Bieber: $\beta = -0.19$, SE = 0.07, $t_{(38)} = 2.89$, $p = 0.003$, 95% CI = $-0.33$ to $-0.06$; Ellen DeGeneres: $\beta = -0.17$, SE = 0.05, $t_{(38)} = 3.19$, $p = 0.001$, 95% CI = $-0.28$ to $-0.07$; Kim Kardashian: $\beta = -0.15$, SE = 0.06, $t_{(38)} = 2.62$, $p = 0.006$, 95% CI = $-0.26$ to $-0.04$) relative to public figures in the domains of politics or business (Barack Obama: $\beta = -0.10$, SE = 0.05, $t_{(38)} = 2.02$, $p = .03$, 95% CI = $-0.21$ to

−0.003; Mark Zuckerberg: $\beta = -0.11$, SE = 0.06, $t_{(38)} = 1.96$, $p = 0.03$, 95% CI = −0.23 to 0.001). Moreover, a post hoc test for moderation by celebrity type confirmed that the strength of the association did in fact differ significantly between pop culture celebrities and public figures in politics or business. Pop culture celebrities (Justin Bieber, Ellen DeGeneres, Kim Kardashian) were coded as 1.0 and public figures in politics or business (Barack Obama, Mark Zuckerberg) were coded as −1.0. Lonelier individuals were significantly more idiosyncratic in their neural representations of pop culture celebrities relative to public figures in politics or business as indicated by a statistically significant interaction between mean loneliness scores and the contrast of pop culture celebrity vs. political/business celebrity: $\beta = -0.03$, SE = 0.01, $t_{(3858)} = 3.33$, $p < 0.001$, 95% CI = −0.06 to −0.01. Importantly, there were no statistically significant associations between loneliness and ratings of familiarity, closeness, or similarity to self for any of the target celebrities (see methods section for details). This suggests that the differences observed between pop culture celebrities and political/business celebrities are unlikely to be due to different levels of knowledge or engagement with these celebrities depending on participants' loneliness. Together, the results in this section demonstrate that lonelier participants exhibited more idiosyncratic neural representations of celebrities in the MPFC compared to those who were less lonely, and, further, that their neural representations were especially idiosyncratic (A) when a strong group consensus existed, and (B) for pop culture celebrities relative to political/business celebrities.

### Study 1 A – similarity to group-consensus neural representations of celebrities

We next tested the hypothesis that lonelier individuals' neural representations of celebrities are especially dissimilar to the group consensus. Whereas the analyses in the previous section tested whether lonelier participants' neural representations of celebrities are more idiosyncratic by examining neural similarity between every possible pair of participants, in this case each individual participant's neural representation of a celebrity was compared to the group-consensus neural representation of that celebrity. Specifically, for each celebrity we computed the weighted average of all neural representations of that celebrity with greater weight being given to participants who were closest to the point of greatest convergence across all participants (see Fig. 1 and methods section for further details). These weighted average neural patterns of activity for each celebrity served as our group-consensus neural representations. Consistent with our hypotheses, results indicated that lonelier participants' neural representations of celebrities in the MPFC were more dissimilar to group-consensus neural representations of celebrities ($\beta = -0.28$, SE = 0.13, $t_{(38)} = 2.07$, $p = 0.02$, 95% CI = −0.54 to −0.02). The bottom portion of Fig. 2 depicts this pattern of results for one of the target celebrities, Ellen DeGeneres. Regarding differences according to type of celebrity, though the strongest associations between loneliness and dissimilarity to group-consensus neural representations were observed for pop culture celebrities (Justin Bieber: $\beta = -0.31$, SE = 0.15, $t_{(38)} = 2.03$, $p = 0.02$, 95% CI = −0.62 to −0.0005; Ellen DeGeneres: $\beta = -0.33$, SE = 0.15, $t_{(38)} = 2.20$, $p = 0.02$, 95% CI = −0.64 to −0.03; Kim Kardashian: $\beta = -0.29$, SE = 0.15, $t_{(38)} = 1.89$, $p = 0.03$, 95% CI = −0.60 to 0.02) relative to public figures in the domains of politics or business (Barack Obama: $\beta = -0.22$, SE = 0.16, $t_{(38)} = 1.40$, $p = 0.08$, 95% CI = −0.54 to 0.10; Mark Zuckerberg: $\beta = -0.22$, SE = 0.16, $t_{(38)} = 1.38$, $p = 0.09$, 95% CI = −0.53 to 0.10), the interaction between loneliness and the contrast of pop culture celebrity vs. political/business celebrity was not statistically significant in this case ($\beta = -0.05$, SE = 0.04, $t_{(158)} = 1.23$, $p = 0.22$, 95% CI = −0.13 to 0.02). We additionally note that the association between loneliness and similarity to the group-consensus neural representations of celebrities in the MPFC remains significant when defined as the unweighted average across participants ($\beta = -0.28$, SE = 0.13, $t_{(38)} = 2.07$, $p = 0.02$, 95% CI = −0.54 to −0.02) or the distance-weighted average across participants ($\beta = -0.28$, SE = 0.13, $t_{(38)} = 2.10$, $p = 0.02$, 95% CI = −0.56 to −0.02). In our view, the weighted average based on proximity to the point of greatest convergence better reflects the concept of the zeitgeist than either of the other two

approaches, but it is useful to know that the result does not depend on this analytic decision.

### Study 1 A – ruling out the potential role of emotional closeness to celebrities

Because previous research has demonstrated that the MPFC represents the interpersonal closeness of others[20,42], and there is some evidence suggesting that lonelier individuals may be more likely to form parasocial attachments to media figures[48,49], we additionally tested whether there was an association between mean ratings of closeness to celebrities and pairwise neural similarity, as well as similarity to group-consensus neural representations. The purpose of these additional analyses was to determine the extent to which the findings related to loneliness were attributable to a tendency to form stronger attachments to media figures or else independent of any possible associations with feelings of closeness. Results indicated that pairs with greater mean ratings of closeness to a celebrity were more dissimilar in their neural representations of that celebrity in the MPFC ($\beta = -0.08$, SE = 0.02, $t_{(480)} = 5.25$, $p < 0.001$, 95% CI = −0.12 to −0.05). Including both mean ratings of closeness to celebrities and mean loneliness scores as predictors of pairwise neural similarity, however, indicated that these two effects were independent of one another (mean loneliness: $\beta = -0.14$, SE = 0.05, $t_{(39)} = 2.84$, $p = 0.004$, 95% CI = −0.23 to −0.04; mean closeness: $\beta = -0.07$, SE = 0.01, $t_{(3610)} = 5.12$, $p < 0.001$, 95% CI = −0.12 to −0.05). It was also the case that participants who reported feeling closer to a celebrity were more dissimilar from the group-consensus neural representation of that celebrity in the MPFC ($\beta = -0.15$, SE = 0.05, $t_{(183)} = 2.70$, $p = 0.008$, 95% CI = −0.25 to −0.04). Including both ratings of closeness to celebrities and loneliness scores as predictors of similarity to the group-consensus neural representations of celebrities, however, again indicated that these two effects were independent of one another (loneliness: $\beta = -0.26$, SE = 0.13, $t_{(38)} = 1.98$, $p = 0.03$, 95% CI = −0.52 to −0.02; closeness: $\beta = -0.14$, SE = 0.05, $t_{(183)} = 2.63$, $p = 0.009$, 95% CI = −0.24 to −0.03).

### Study 1B – pairwise similarity in neural representations of celebrities

In Study 1B, we attempted to replicate the statistically significant results found in Study 1 A using data from 40 participants from an independent dataset in which participants completed the same trait evaluation task for the same set of five celebrities. We refined our focus in line with the results of Study 1 A. First, because in Study 1 A the predicted association between loneliness and neural representational similarity was observed in the MPFC but not the PC/PCC, we focused only on the former in Study 1B. Second, because the pairwise analyses in Study 1 A were significantly stronger for pop culture celebrities relative to public figures in politics or business, we additionally focused our replication attempt only on the celebrities that fit this category: Ellen DeGeneres, Kim Kardashian, and Justin Bieber.

Consistent with the results of Study 1 A, pairs with greater mean loneliness scores were more dissimilar in their neural representations of pop culture celebrities in the MPFC ($\beta = -0.17$, SE = 0.06, $t_{(38)} = 2.68$, $p = 0.005$, 95% CI = −0.28 to −0.04). In the Study 1B dataset, there were no statistically significant differences in the distributions of pairwise neural similarity between the three pop culture celebrities (Kim Kardashian vs. Ellen DeGeneres: $\beta = 0.01$, SE = 0.01, $t_{(1520)} = 0.61$, $p = 0.54$, 95% CI = −0.01 to 0.02; Kim Kardashian vs. Justin Bieber: $\beta = 0.01$, SE = 0.01, $t_{(1520)} = 0.97$, $p = 0.33$, 95% CI = −0.01 to 0.03; Ellen DeGeneres vs. Justin Bieber: $\beta = 0.003$, SE = 0.01, $t_{(1520)} = 0.37$, $p = 0.71$, 95% CI = −0.02 to 0.02). In other words, all three celebrities showed similar levels of neural similarity across all pairs. We therefore did not test for moderation by level of consensus in this dataset.

### Study 1B – similarity to group-consensus neural representations of celebrities

As in Study 1 A, we next computed the group-consensus neural representation for each celebrity, i.e., the weighted average of all neural representations with greater weight being given to participants who were closest

to the point of greatest convergence across all participants (see Fig. 1 and methods section for further details). We again observed that lonelier participants' neural representations of pop culture celebrities in the MPFC were more dissimilar to group-consensus neural representations ($\beta = -0.30$, SE = 0.15, $t_{(38)} = 2.10$, $p = 0.02$, 95% CI = $-0.58$ to $-0.03$). We additionally note that the association between loneliness and similarity to the group-consensus neural representations of celebrities in the MPFC remains statistically significant when defined as the unweighted average across participants ($\beta = -0.28$, SE = 0.15, $t_{(38)} = 1.88$, $p = 0.03$, 95% CI = $-0.57$ to 0.01) or the distance-weighted average across participants ($\beta = -0.31$, SE = 0.14, $t_{(38)} = 2.11$, $p = 0.02$, 95% CI = $-0.57$ to $-0.02$).

### Study 1B – Ruling out the potential role of emotional closeness to celebrities

In contrast to Study 1 A, in Study 1B there were no significant associations between ratings of closeness to celebrities and more idiosyncratic neural representations of them. There was no statistically significant association between pairs' mean closeness scores and pairwise neural similarity in the MPFC ($\beta = 0.05$, SE = 0.03, $t_{(801)} = 1.59$, $p = 0.11$, 95% CI = $-0.02$ to 0.11) and mean loneliness scores remained a significant predictor controlling for mean closeness scores (mean loneliness: $\beta = -0.16$, SE = 0.06, $t_{(36)} = 2.60$, $p = 0.007$, 95% CI = $-0.28$ to $-0.03$; mean closeness: $\beta = 0.05$, SE = 0.03, $t_{(732)} = 1.57$, $p = 0.12$, 95% CI = $-0.01$ to 0.11). There was no statistically significant association between individuals' closeness scores and their similarity to the group-consensus neural representations of celebrities in the MPFC ($\beta = 0.03$, SE = 0.09, $t_{(116)} = 0.35$, $p = 0.73$, 95% CI = $-0.15$ to 0.19) and loneliness remained a significant predictor controlling for ratings of closeness (loneliness: $\beta = -0.31$, SE = 0.15, $t_{(35)} = 2.08$, $p = 0.02$, 95% CI = $-0.59$ to $-0.03$; closeness: $\beta = 0.03$, SE = 0.09, $t_{(116)} = 0.34$, $p = 0.74$, 95% CI = $-0.13$ to 0.21). This again suggests that observed differences in neural similarity as a function of loneliness are not driven by or redundant with differences in emotional closeness to celebrities.

### Study 1B – controlling for participant behavior during the fMRI trait-rating task

In Study 1B, we additionally examined and controlled for participants' behavior during the fMRI trait-rating task, first by modeling similarity in pairs of participants' ratings of each celebrity's traits. We tested whether pairwise similarity in trait ratings of celebrities exhibited the same Anna Karenina effect with regards to loneliness as was observed for pairwise neural similarity in the MPFC. Results indicated that there was no statistically significant association between mean loneliness and pairwise similarity in trait ratings of celebrities: $\beta = -0.01$, SE = 0.07, $t_{(38)} = 0.12$, $p = 0.90$, 95% CI = $-0.14$ to 0.13, BF10 = 0.04. Further, when including both mean loneliness and pairwise similarity in trait ratings of celebrities in a model predicting pairwise similarity in neural representations of celebrities in the MPFC, mean loneliness remained a significant predictor (mean loneliness: $\beta = -0.17$, SE = 0.06, $t_{(38)} = 2.68$, $p = 0.005$, 95% CI = $-0.28$ to $-0.06$; pairwise similarity in trait ratings: $\beta = -0.005$, SE = 0.02, $t_{(2334)} = 0.31$, $p = 0.76$, 95% CI = $-0.03$ to 0.03).

Next, we examined individual participants' similarity to the group-consensus trait perceptions of celebrities. Results indicated that there was no statistically significant association between loneliness and similarity to group-consensus perceptions of celebrities' traits ($\beta = 0.09$, SE = 0.14, $t_{(38)} = 0.65$, $p = 0.52$, 95% CI = $-0.18$ to 0.38, BF10 = 0.31). Further, when including both loneliness scores and similarity to group-consensus perceptions of celebrities' traits in a model predicting similarity to group-consensus neural representations of celebrities in the MPFC, loneliness remained a significant predictor (loneliness: $\beta = -0.29$, SE = 0.15, $t_{(38)} = 1.94$, $p = 0.03$, 95% CI = $-0.56$ to $-0.001$; similarity to group-consensus trait perceptions: $\beta = -0.19$, SE = 0.07, $t_{(107)} = 2.71$, $p = 0.008$, 95% CI = $-0.33$ to $-0.05$).

Next, we examined whether the number of missing responses or average response time (RT) was significantly associated with participants' loneliness scores, which could reflect different levels of engagement with the task or confidence in one's responses. While there was no statistically significant evidence for an association between loneliness and participants' average RT ($\beta = -0.21$, SE = 0.15, $t_{(37)} = 1.34$, $p = 0.19$, 95% CI = $-0.51$ to 0.09, BF10 = 1.91), there was a statistically significant negative association between loneliness and the number of responses participants missed ($\beta = -0.30$, SE = 0.14, $t_{(38)} = 2.11$, $p = 0.04$, 95% CI = $-0.60$ to $-0.02$) such that lonelier participants missed fewer responses while completing the trait-rating task in the scanner. However, when including mean loneliness, pairwise similarity in number of missing responses, and pairwise similarity in average RT in a model predicting pairwise similarity in neural representations of celebrities in the MPFC, mean loneliness remained a significant predictor (mean loneliness: $\beta = -0.15$, SE = 0.07, $t_{(37)} = 2.37$, $p = 0.01$, 95% CI = $-0.28$ to $-0.03$; pairwise similarity in number of missing responses: $\beta = -0.03$, SE = 0.02, $t_{(2209)} = 1.89$, $p = 0.06$, 95% CI = $-0.06$ to 0.003; pairwise similarity in average RT: $\beta = 0.02$, SE = 0.02, $t_{(2177)} = 0.95$, $p = 0.34$, 95% CI = $-0.02$ to 0.05). Similarly, when including loneliness, number of missing responses, and average RT in a model predicting similarity to group-consensus neural representations of celebrities in the MPFC, loneliness remained a significant predictor (loneliness: $\beta = -0.26$, SE = 0.15, $t_{(38)} = 1.74$, $p = 0.045$, 95% CI = $-0.55$ to $-0.003$; number of missing responses: $\beta = -0.10$, SE = 0.09, $t_{(109)} = 1.16$, $p = 0.25$, 95% CI = $-0.27$ to 0.06; average RT: $\beta = 0.25$, SE = 0.10, $t_{(111)} = 2.43$, $p = 0.02$, 95% CI = 0.06 to 0.44).

The results in this section indicate the association between loneliness and idiosyncratic neural representations is robust to nuances of the task: results hold after controlling for participants' trait ratings, the number of responses participants missed, and how quickly they made their trait judgments. This suggests that observed differences in neural similarity as a function of loneliness are not driven by or redundant with differences in attention to the task.

### Study 2 – pairwise semantic similarity

Studies 1 A and 1B demonstrated that lonelier individuals have more idiosyncratic neural representations of prominent celebrities. The pairwise analyses in Study 1 A, moreover, suggested this was particularly true for famous figures from pop culture for whom there is a strong group consensus. Given the important role of communication in creating a shared reality[17–19,53,54], in Study 2 ($N = 923$) we tested whether a similar pattern might be observed when lonely individuals communicate their perceptions of celebrities to others —particularly when communicating about pop culture celebrities for which there is a strong group consensus. Participants wrote a paragraph describing a celebrity they indicated they were familiar with as if to a friend who had never heard of that person. With the data from these free-response paragraphs we conducted a text analysis using Google's Universal Sentence Encoder (USE)[50] to compute the semantic similarity between each pair of participants' descriptions of a celebrity.

Because we included more celebrity stimuli in Study 2, we were well-equipped to test whether any association between loneliness and semantic similarity was moderated by the strength of consensus in the language participants used to describe a given celebrity, which we did in two ways. First, we used a median split to divide the ten pop culture celebrities chosen for the study into two groups based on the mean pairwise semantic similarity for each celebrity: high-consensus celebrities (Ellen DeGeneres, Kim Kardashian, Harrison Ford, Michael Jordan, and Keanu Reeves) and low-consensus celebrities (Justin Bieber, Cameron Diaz, Megan Fox, Will Smith, and Justin Timberlake). Second, because there was greater variability in consensus across all the celebrity targets (19 of 45 pairs exhibited significantly different levels of consensus, see Supplementary Table 1 for full results), we were also well equipped to test moderation by level of consensus by modeling it continuously (i.e., as each celebrity's mean pairwise semantic similarity).

As shown in Fig. 3, there was a statistically significant interaction between mean loneliness and the contrast of high vs. low consensus celebrities ($\beta = -0.04$, SE = 0.02, $t_{(915)} = 2.18$, $p = 0.01$, 95% CI = $-0.08$ to $-0.004$). This interaction was also statistically significant when modeling

consensus in a continuous fashion ($\beta = -0.04$, SE = 0.02, $t_{(917)} = 2.19$, $p = 0.01$, 95% CI = $-0.08$ to $-0.005$). A follow-up analysis including data only from high-consensus celebrities confirmed that under the condition of a strong general consensus there was in fact a statistically significant negative association between mean loneliness and pairwise semantic similarity ($\beta = -0.07$, SE = 0.03, $t_{(445)} = 2.41$, $p = 0.008$, 95% CI = $-0.12$ to $-0.02$). Further, this result held after controlling for other variables (e.g., age, liking, similarity in trait ratings, word count) that also predicted pairwise semantic similarity (see Supplementary Information: Supplementary Results pages 4–5 and Supplementary Tables 2, 3). (We also note here that though similarity in word count was associated with pairwise semantic similarity, there was no statistically significant association between loneliness and word count: $\beta = 0.02$, SE = 0.03, $t_{(917)} = 0.74$, $p = .46$, 95% CI = $-0.05$ to 0.09.) It is noteworthy that in this study pairwise effects seem relatively specific to the high-consensus celebrities. When looking across all celebrities, regardless of consensus level, there was also a negative association between mean loneliness scores and pairwise semantic similarity, but the effect was only marginally statistically significant in this case (i.e., it was attenuated by the addition of low-consensus celebrities; $\beta = -0.03$, SE = 0.02, $t_{(914)} = 1.45$, $p = 0.07$, 95% CI = $-0.07$ to 0.01).

### Study 2 – similarity to group-consensus semantic representations of celebrities

We next computed a group-consensus vector of semantic features for each of the five high-consensus celebrities and then tested whether lonelier participants tended to be especially dissimilar to the weighted-average group consensus. Results indicated that for high-consensus celebrities, lonelier individuals were more dissimilar to the group consensus in terms of the language they used to describe them ($\beta = -0.08$, SE = 0.05, $t_{(445)} = 1.79$, $p = 0.04$, 95% CI = $-0.18$ to $-0.001$). We further note that this association remained statistically significant when defining the group consensus as the unweighted average across participants ($\beta = -0.08$, SE = 0.05, $t_{(445)} = 1.73$, $p = 0.04$, 95% CI = $-0.18$ to 0.01) or the distance-weighted average across participants ($\beta = -0.08$, SE = 0.05, $t_{(445)} = 1.74$, $p = 0.04$, 95% CI = $-0.17$ to 0.003). Race, similarity to group-consensus perceptions of celebrities' traits, and word count were also significantly associated with similarity to the group-consensus semantic representations of high-consensus celebrities (see Supplementary Table 4). However, when simultaneously controlling for these three variables, loneliness remained a significant predictor of similarity to group-consensus semantic representations of high-consensus celebrities (see Supplementary Information: Supplementary Results page 5).

Finally, the association between loneliness and idiosyncratic representations of celebrities in Study 2 was specific to semantic representations. Although there was a statistically significant association between trait representations of celebrities and semantic representations of celebrities, there was no statistically significant evidence that lonelier individuals were more idiosyncratic in their perceptions of celebrities' traits relative to less lonely individuals (see Supplementary Information: Supplementary Results pages 5–6).

### Study 2 – subjective overlap and accuracy in perceptions of celebrities

Participants were also asked to rate, on a 0–100 scale, the extent to which they believed their perception of a celebrity was accurate and shared by other people in their social circles (order of written description and ratings randomized across participants). One typical characteristic of loneliness is the perception that one's ideas are not shared by those around them[1]. Our results suggest that this common subjective experience for lonelier individuals extends even to perceptions of pop culture celebrities. Specifically, greater loneliness was statistically significantly associated with decreased feelings of one's perception of a famous celebrity being similar to those of the people around them (e.g., "friends, family, co-workers, etc."): $\beta = -0.09$, SE = 0.03, $t_{(915)} = 2.65$, $p = 0.008$, 95% CI = $-0.15$ to $-0.02$). Further, consistent with theorizing regarding shared reality, which suggests that a perceived lack of shared reality with others should be associated with decreased

certainty in one's knowledge[19,54], greater loneliness was also statistically significantly associated with decreased feelings of one's perception of a famous celebrity being accurate ($\beta = -0.11$, SE = 0.03, $t_{(915)} = 3.48$, $p < 0.001$, 95% CI = $-0.18$ to $-0.05$). Importantly, there were no statistically significant associations between loneliness and ratings of familiarity ($\beta = -0.05$, SE = 0.03, $t_{(921)} = 1.61$, $p = 0.11$, 95% CI = $-0.12$ to 0.01), liking ($\beta = -0.05$, SE = 0.03, $t_{(921)} = 1.61$, $p = 0.11$, 95% CI = $-0.12$ to 0.01), closeness ($\beta = 0.01$, SE = 0.03, $t_{(921)} = 0.24$, $p = .81$, 95% CI = $-0.06$ to 0.07), or similarity to self ($\beta = -0.03$, SE = 0.03, $t_{(921)} = 0.80$, $p = 0.43$, 95% CI = $-0.09$ to 0.04). This suggests that the differences observed in subjective perceptions of one's impression of a celebrity being shared by others and accurate is likely not due to differential levels of knowledge or engagement with these celebrities depending on participants' loneliness.

Lonelier participants in Study 2 were more likely to report feeling that their perceptions of celebrities were not accurate and not shared by those around them. More importantly, Study 2 elucidated the conditions under which this subjective sense is in fact true by focusing on a fundamental tool for building shared reality with others: verbal communication. The results of Study 2 provide a behavioral replication and extension of the neural findings in Study 1, demonstrating that under conditions of high group consensus, when lonelier individuals communicate their perceptions of celebrities, the language they use to describe them is idiosyncratic in meaning compared to less lonely individuals.

## Discussion

Loneliness is a widespread epidemic in contemporary life[55]. It was already on the rise before 2020[56,57] and has only been exacerbated by the COVID-19 pandemic[58–60]. It is associated with a host of negative outcomes, including increased risk of depression and mortality[55,61]. Loneliness is characterized by feelings of not being understood by others and not having ideas or interests in common with those in one's social network[1,2]. Here we demonstrate that there is ground truth to this perception. Lonely individuals' mental representations of contemporary cultural figures (i.e., well-known celebrities) stray from the group-consensus representation. We observed this in two ways and across two different fMRI datasets: lonelier individuals' neural representations in the MPFC differed significantly from others', especially when a strong consensus existed for a celebrity, and they deviated from group-consensus representations. Study 2 reaffirmed these findings behaviorally. When a strong group consensus is present for a celebrity, lonelier individuals use idiosyncratic language when expressing their perceptions of them. The results demonstrate that loneliness corresponds with idiosyncratic knowledge that deviates from the zeitgeist.

The results add to a growing body of work on the brain basis of loneliness, which collectively implicate the brain's default mode network or default network in social isolation. Two prior studies are particularly relevant to the present work. One study found subjective perceptions of loneliness[7] and the other found an objective lack of social ties[6] are associated with idiosyncratic neural activity while viewing popular media (e.g., video footage from *America's Funniest Home Videos*). In both prior studies, default network regions outside of MPFC demonstrated idiosyncratic responding and the only default network region implicated in both prior studies was the dorsomedial prefrontal cortex (DMPFC). The DMPFC is reliably associated with social cognition, particularly abstract social knowledge[62–65], inferences about dissimilar others[66], and shared social schemas[14,67]. In contrast, we observed lonelier participants' neural representations of celebrities diverged in the MPFC, which is more associated with well-known, self-relevant, and crystalized social knowledge about specific individuals[16,20,42,43,68]. Loneliness is also associated with increased integrity of the fornix pathway[69] by which the MPFC receives signals from the hippocampus, a brain region reliably associated with memory formation. In fact, as new episodic memories transform into semantic knowledge, they tend to move from the hippocampus to cortical regions including the MPFC[70]. Deviating from well-established cultural knowledge may therefore reflect a longer-term process that begins in the DMPFC (during social inferences made on the fly when witnessing new information), followed by

short-term storage in the hippocampus and eventual semantic knowledge about individual people in the MPFC. More broadly, our loneliness-idiosyncrasy results in the MPFC, in conjunction with prior findings in the DMPFC[6,7], suggest loneliness may be associated with idiosyncratic social cognition in multiple ways distributed throughout default network regions. Future work is required to fully map the links between loneliness and idiosyncratic processing.

The findings presented here are correlational and therefore the direction of causality is unknown. Below we speculate on the possible ways (1) deviating from the zeitgeist may induce loneliness, (2) loneliness may induce deviating from the zeitgeist, and (3) how the relationship between these two variables may compound loneliness in a cyclical fashion.

How would idiosyncratic representations of shared cultural knowledge induce loneliness? Deviating from the zeitgeist should make it more difficult for individuals to quickly relate to and connect with others. Shared reality theory proposes that people are motivated to perceive that their inner states match those of others, in part, to feel socially connected to one another[19,54]. The fundamental tool for establishing shared reality with others is verbal communication[18,19,53,54], and the results of Study 2 suggest that lonelier individuals may start at a disadvantage when it comes to collaboratively constructing consensus with others, particularly when a consensus already appears to exist.

Deviating from the zeitgeist may also induce loneliness if people realize their views do not align with the norm—this may signal to them that they do not fit in. In fact, the stronger the consensus surrounding a given viewpoint, the stronger the cue that those who do not hold that viewpoint do not belong. Consistent with this idea, we found that celebrities with the strongest consensus were the ones lonelier participants represented the most idiosyncratically. This was true both with respect to neural representations of celebrities in the MPFC (Study 1 A) and in terms of semantic similarity when describing celebrities (Study 2). Individuals who chronically feel as though their ideas are not shared by others might be especially likely to fixate on the ways in which their impressions differ from others', and perceptions of a strong consensus should make these contrasts, and their implications, particularly salient.

On the flipside, could feeling lonely generate idiosyncratic representations of otherwise shared cultural knowledge? Although acute loneliness motivates social interaction[71–73], chronic loneliness corresponds with social withdrawal[74]. Loneliness is also associated with internally-focused thinking removed from "the here and now" social environment, such as reminiscing about the past[75] and imagining social interactions, including with non-human others[76]. This increase in internally focused thinking is theorized to be an attempt (not necessarily conscious) by lonely individuals to compensate for their perceived lack of meaningful connection with members of their real-life social networks[48,69]. One proposed function of the default network is to integrate incoming extrinsic information with existing intrinsic information in order to make sense of the world[77]. This integration process is key to establishing shared reality with others and in turn building communities and networks. While any individual's intrinsic information (e.g., knowledge and memories) is inherently idiosyncratic, if one becomes socially withdrawn and/or too deeply entrenched in their internal milieu it could over time undermine this integration process, bending incoming information to fit one's (overly) unique perspective rather than facilitating achieving common understandings with others. This possibility is supported by the results of Study 2, which demonstrated that lonelier individuals were more idiosyncratic in the way they described pop culture celebrities. In other words, when tasked with communicating social information to others, the language used by lonelier individuals did not reflect the group's shared understanding but rather their own unique take. Additional evidence for this internal focus comes from work demonstrating that when faced with images of people in unpleasant circumstances, lonelier individuals exhibit less activity in the temporoparietal junction (TPJ)[78], a region strongly implicated in inferring the mental states of others[79]. This suggests that in contexts that tend to reorient most people's attention toward the experiences of others, lonelier individuals may remain focused on their own internal experience[80].

A third possibility is that the relationship between loneliness and deviating from the zeitgeist is not unidirectional and instead a cyclical process. Loneliness is known to build on itself: lonely individuals frequently engage in behaviors that exacerbate their loneliness[80] and loneliness can even spread through social networks[81]. Yet, the underlying mechanisms by which loneliness begets more loneliness remain to be determined. While multiple mechanisms are likely involved, our results hint at one possible route. Shared reality with others facilitates connection not only by establishing but also by reinforcing a sense of shared norms[19,54]. Straying from the zeitgeist puts people at a disadvantage in finding common ground in everyday social life, which may circle back and intensify feelings of isolation.

A cyclical relationship may also stem from the epistemic goals met through shared reality[19,54]. If one perceives that another person's impressions agree with their own, this corroborates that impression and increases one's confidence in it. Consistent with this view, the results of Study 2 demonstrated that lonelier individuals perceived that their impressions of famous celebrities were not shared with others and were less accurate. A potential consequence of idiosyncratic representations of culturally significant figures, therefore, is not just a decreased sense of connection to others but also decreased confidence in one's own knowledge and impressions. This uncertainty may make lonely individuals less likely to volunteer their own opinions and engage in the types of consensus-building conversations that can lead to a sense of shared reality, providing yet another avenue through which loneliness can be self-reinforcing.

Should lonelier individuals simply strive to adhere to the zeitgeist, then? We believe such a conclusion would be misguided. Shared perceptions of prominent cultural figures are not necessarily more accurate or beneficial to society just by virtue of being more common. Future research into the neural correlates of shared reality could instead seek to build on work demonstrating the conditions under which a consensus is reached when there are many disparate viewpoints[53], with a particular focus on what conditions promote consideration of peripheral perspectives. For example, recent work shows that interventions aimed at increasing the motivation to be empathetic can lead to reduced loneliness within the targeted social network[82], suggesting those at the periphery can be brought into the fold when people are willing to expend the prosocial energy to do so. Future work can aim to uncover the ways in which isolated individuals can be made to feel their reality is shared with those around them; crucially, in a way that does not always value the dominant viewpoint above less well-represented ones.

Loneliness is a multifaceted phenomenon that likely has many causes, including societal and technological ones[83]. For example, there is evidence that loneliness is more prevalent among those with lower socioeconomic status[84] as well as those who self-report social media overuse[85]. Any account of loneliness is incomplete without considering the independent contribution of these systemic and behavioral factors. Though we did not account for these factors in the present work, it is possible they might even directly contribute to the association observed between loneliness and idiosyncratic representations of celebrities. Disadvantaged social groups face barriers to resources, including informational resources, that could lead to representations of social knowledge that differ from the perceived norm. Overuse of social media, moreover, may lead to social knowledge that differs from the perceived norm if the sources of one's information are unusual, or obscure compared to those of others in one's real-life social network. Future work can determine the extent to which sociodemographic and behavioral factors contribute to the link between loneliness and idiosyncrasy.

## Limitations
We assessed trait loneliness, which captures chronic feelings of disconnection. However, loneliness can also be a transitory state. Additional work is needed to determine if even brief, temporary moments of loneliness relate to mental representations that deviate from the norm.

The present study also focused on mental representations of prominent, culturally significant celebrities. Whether the idiosyncratic representations of celebrities exhibited by lonelier individuals generalize to representations of other classes of shared cultural knowledge is unknown, as

is whether the results generalize to representations of all celebrities. In our view, celebrities are a good starting point to test the possibility that lonely individuals' mental representations deviate from the zeitgeist. Celebrities are extensively showcased in news media[86] and the most prominent ones are among the highest sought out topics on sites like Wikipedia[3]. Prior work even finds that celebrities that generate common ground between strangers are disproportionately discussed in conversation[11], suggesting shared celebrity knowledge can provide a "foot in the door" to forming ties with others. That said, celebrities are not the only components of the zeitgeist. Future work is needed to determine whether other forms of shared cultural knowledge are also idiosyncratic in lonelier individuals. Interestingly, in Study 1 A the neural results for the pairwise analyses were stronger for pop culture celebrities relative to political and business celebrities, although this was observed in a sample of young adults. It is possible that across the lifespan different aspects of the zeitgeist, such as political perspectives and social values, change in their relevance to loneliness.

As can be seen in Fig. 2, participants low in loneliness tended to be tightly clustered together while participants higher in loneliness were more variable with some being highly unique while others were fairly similar to the less lonely participants. This suggests that while there is a general trend toward more idiosyncratic representations of culturally significant figures for lonelier individuals, this pattern does not hold true for every individual struggling with social isolation. It is also notable that the fMRI experiments prioritized having a large number of trials per celebrity to get reliable neural estimates over a large number of celebrities. Consequently, a limitation of the present study is the small number of celebrity targets especially with regard to the small number of celebrities for the categories examined (i.e., pop culture vs. political/business celebrity and high vs. low-consensus celebrity). It is therefore unclear if the MPFC findings would generalize to the wider population of celebrities. Given that the results were conceptually replicated in Study 2 with a larger number of celebrities, we predict that the neural results would generalize to other well-known celebrities. Future work can identify what variables distinguish between lonely individuals who exhibit idiosyncratic representations of culturally significant figures and ones who do not, as well as whether findings generalize across multiple celebrities.

Finally, another limitation of the present study is the operationalization of neural representations of celebrities using a single experimental task. Future work could better elucidate the full range of social cognitive knowledge that is represented idiosyncratically by lonelier individuals by using other (or multiple other) social cognitive tasks that have been implemented in social neuroscience fMRI studies, such as mentalizing[3] or mental simulation tasks[87,88]. Relatedly, future work may seek to determine whether the semantic results reported here emerge under a more diverse set of operationalizations, including more naturalistic circumstances (e.g., data from actual conversations).

## Conclusion

Shared reality fosters social connections between people and increases confidence in one's knowledge because it is corroborated by others. While lonely individuals report feeling disconnected from others in terms of their interests and ideas it was previously unclear to what extent this is true with respect to the zeitgeist—defined here as the widely shared perceptions between members of contemporary culture. Our findings provide evidence that loneliness is associated with deviations from the zeitgeist, specifically when it comes to perceptions of well-known celebrities. Loneliness corresponded with idiosyncratic neural representations of celebrities as well as more idiosyncratic communication about celebrities, particularly when an otherwise strong consensus existed between less lonely people. Lonely individuals' feeling that their ideas are not shared by the people around them is more than metaphorical; it is objectively reflected in idiosyncratic knowledge of contemporary culture that strays from the consensus.

## Data availability

Data from all studies are freely available online on the Open Science Framework (https://osf.io/v8wz5/).

## Code availability

Code from all studies are freely available online on the Open Science Framework (https://osf.io/v8wz5/).

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

## Acknowledgements
The authors would like to thank E. Tory Higgins and Emma Miller for their comments on this manuscript. This work was supported by NIMH grant R01MH125406 to MLM. The funder had no role in the conceptualization, design, data collection, analysis, decision to publish, or preparation of this manuscript.

## Author contributions
A.L.C. and M.L.M. designed Study 1 A. ALC collected the data for Study 1 A. All authors contributed to the design of Study 1B. S.I. and T.W.B. collected the data for Study 1B. T.W.B. developed the hypotheses and conducted all analyses for Studies 1 A and 1B. T.W.B., S.I. and M.L.M. designed Study 2. S.I. collected the data for Study 2. T.W.B. and S.I. conducted the analyses for Study 2. T.W.B. and M.L.M. wrote the initial draft of the manuscript. All authors commented on and approved all drafts of the manuscript. M.L.M. supervised all the work described herein.

## Competing interests
The authors declare no competing interests.
