## [Peer Review File · Communications Psychology]

This manuscript has been previously reviewed at another Nature Portfolio journal. This document only contains reviewer comments and rebuttal letters for versions considered at Communications Psychology.

8th Jan 24

Dear Dr Broom,

Thank you for your patience during the peer-review process. Your manuscript titled "Keeping up with others' perceptions of the Kardashians: Lonely individuals' neural representations and language use do not reflect the zeitgeist" has now been seen by 2 of the original reviewers, and I include their comments at the end of this message. They find your work of interest, but raised some important points. We are interested in the possibility of publishing your study in *Communications Psychology*, but would like to consider your responses to these concerns and assess a revised manuscript before we make a final decision on publication.

We therefore invite you to revise and resubmit your manuscript, along with a point-by-point response to the reviewers. Please highlight all changes in the manuscript text file.

Our policy on statistics reporting and interpretation is that non-significant results from NHST may not be interpreted. This means that the absence of evidence for an effect or a difference may not be treated as evidence for the absence of an effect or difference. To align with this policy, please add Bayes factors for the control analyses in Studies 1a and 1b to provide positive evidence for the null. If you obtain $BFs < 3$ in favour of the null hypothesis, you must change the interpretation accordingly.

Along the same lines, the (lack of) association between MPFC representational similarity and similarity of trait evaluations may not be interpreted. We advise simply stating the finding without interpretation.

Presentationally, we ask you to avoid altering the hypotheses. Instead, please discuss other possible systematic causes influencing your findings (with appropriate citations) but it should be clear that you did not account for these in your study.

Likewise, please do not remove any planned analyses. Limitation of the analyses and operationalization of the constructs should be discussed.

We advise a more nuanced interpretation of the findings regarding the difference in association strength observed between politicians and pop culture celebrities as these differences were not found in all analyses.

I have included a checklist to help ensure your revision aligns with our formatting and reporting requirements.

Please use the following link to submit your revised manuscript, point-by-point response to the referees' comments (which should be in a separate document to any cover letter) and the completed checklist:

[Link redacted]

Please do not hesitate to contact me if you have any questions or would like to discuss these revisions further. We look forward to seeing the revised manuscript and thank you for the opportunity to review your work.

Best regards,

Jennifer Bellingtier

Jennifer Bellingtier, PhD
Senior Editor
Communications Psychology

* TRANSPARENT PEER REVIEW: Communications Psychology uses a transparent peer review system. This means that we publish the editorial decision letters including Reviewers' comments to the authors and the author rebuttal letters online as a supplementary peer review file. However, on author request, confidential information and data can be removed from the published reviewer reports and rebuttal letters prior to publication. If your manuscript has been previously reviewed at another journal, those Reviewers' comments would not form part of the published peer review file.

REVIEWERS' EXPERTISE:

Reviewer #2 social neuroscience / connectedness
Reviewer #3 social neuroscience / connectedness

REVIEWERS' COMMENTS:

Reviewer #2 (Remarks to the Author):

The authors have addressed my concerns, and I am happy to recommend the manuscript for publication.

Reviewer #3 (Remarks to the Author):

The authors have addressed many of my concerns, but a few remain:

1. Conceptual issue: Possible causes of loneliness.

I am happy to see the added discussion of systemic causes of loneliness. However, this is included as a "limitation," which is not quite right. Systemic causes should be discussed alongside other hypothesized causes, such as excessive internally focused thinking.

2. Conceptual issue: Behavior and mental representations of lonely people

The authors report a new analysis controlling for the reaction time and missed responses. Lonely participants missed fewer responses, suggesting differences in attention. But, controlling for these differences, the effect of loneliness on representational similarity persists. This is a nice analysis.

The authors' response describes new analyses that show no correspondence between MPFC representational similarity and similarity of trait evaluations. This indicates that the trait evaluations do not correspond to the content of the MPFC representations (or that fMRI lacks the resolution required to identify the correspondences), and raises the question of what MPFC represents during the reported task. This should be reported and discussed in the manuscript.

The authors' response states that the results "cannot be explained by differences in perceptions of [...] traits." I would amend this to read "measured traits," as it is possible that the MPFC represents traits, just not the ones that were measured.

In lines 648-661 of the revised manuscript, the authors make several questionable claims that seem to spin the mismatch between trait judgments and MPFC results as a kind of robustness check. I think this is a misguided line of argument. Instead, I suggest the authors clearly state that the mismatch between the trait judgments and the MPFC representations is a shortcoming of the paper, while also clearly stating that their main arguments still stand. Social psychology has long had a problem where less than perfect results are wrongly rejected even when the big picture holds up just fine, and I can see why the authors would take a defensive pose, but it is not necessary here.

3. Scope of generalization

The authors' response that celebrities were selected on the basis of popularity is simply not germane to the question of generalization scope. I would strongly encourage the authors to review Clark (1973) and other work on the stimulus-as-fixed-effect fallacy, e.g., Westfall, Nichols, & Yarkoni (2016). That said, the caveat added on line 610 is reasonable, although the authors should add that it is also not clear whether the results will generalize to other celebrities.

I appreciate that the tests using celebrity categories are now identified as post hoc. But I would like to see the small number of celebrities in each category to be directly identified as a limitation in the discussion section. This would be a natural addition to the paragraph starting on line 636.

Line 642: I suggest changing "it is unclear if the MPFC findings would generalize to several celebrities" to "it is unclear if the MPFC findings would generalize to the wider population of celebrities."

Line 73: I suggest changing "logical" to "reasonable."

4. Justification of celebrity trait ID paradigm

The authors note that the UCLA loneliness scale is well-validated, which is why they use overall loneliness rather than the specific relevant survey items. This is all well and good, but it is not germane to the concern that the jump from loneliness to celebrities is poorly justified.

Happily, the authors' reorganization of the introduction fully addresses my concern.

5. Split by level of consensus

I am delighted to see the improvements to Study 2.

The authors note that in Study 1, where Justin Bieber was the only high-consensus celebrity, representing consensus continuously eliminated the effect. This is consistent with the concern that generalizing from small numbers of celebrities (in this case a single celebrity) is not appropriate. The authors note that this result may be driven by "limited variability in consensus across targets," which seems an understatement given that one category consists only of a single target, Justin Bieber. I think the paper would be stronger if this analysis were simply removed, as Study 2 now makes the same point but much more convincingly.

Relatedly, I do not find the authors' claim that "the interaction between loneliness and level of consensus was consistent across two different datasets" compelling, because the authors acknowledge that the effect in one of the datasets disappears when consensus is evaluated as a continuous variable. Why grasp at straws when the improvements to Study 2 are so strong?

The authors note that Justin Bieber's shift from high-consensus in Study 1 to low-consensus in Study 2 may reflect "differences in how the target individual was generally perceived across two different points in time and/or groups of people," arguing that this "could be considered a strength of our findings." This is an interesting possibility, but it is an empirical matter and the authors provide no supporting data or analysis.

6. Distance-weighted average

I am happy to see that results using a distance-weighted mean without MDS are nearly identical to the reported results. I am also happy to see that the results hold without any distance-weighting at all. So why continue to report the MDS results? MDS throws information away, biasing the mean. It may not make a meaningful difference here, but it may make a difference for future studies by other authors that use this one as a template. Even the authors in their revised manuscript make no mention of MDS when justifying this analysis (line 114). Simply using the distance-weighted mean would also resolve the concern that the Gaussian kernel parameters were set arbitrarily.

In their response, the authors argue their approach would better capture multi-modal distributions of representations (i.e., more than one consensus). But this does not match the reported methods, as it would require identifying at least two centroids using at least two Gaussian kernels and a further process to optimize such a model. In any case, while a Gaussian mixture model approach is justifiable, it also would not require MDS.

Refs

Clark, H. H. (1973). The language-as-fixed-effect fallacy: A critique of language statistics in psychological research. *Journal of verbal learning and verbal behavior*, 12(4), 335-359.

Westfall, J., Nichols, T. E., & Yarkoni, T. (2016). Fixing the stimulus-as-fixed-effect fallacy in task fMRI. *Wellcome open research*, 1(23).

REVIEWERS' COMMENTS:

Reviewer #2 (Remarks to the Author):

The authors have addressed my concerns, and I am happy to recommend the manuscript for publication.

Response: *Thank you for your thoughtful consideration of our manuscript. We are glad to hear that we were able to address the concerns you previously raised.*

Reviewer #3 (Remarks to the Author):

The authors have addressed many of my concerns, but a few remain:

1. Conceptual issue: Possible causes of loneliness.

I am happy to see the added discussion of systemic causes of loneliness. However, this is included as a "limitation," which is not quite right. Systemic causes should be discussed alongside other hypothesized causes, such as excessive internally focused thinking.

Response: *We appreciate your point and in line with your recommendation we have moved our discussion of systemic causes of loneliness out of the limitations section and instead discuss it alongside the other hypothesized causes we consider (see page 48-49, lines 1095-1107 of main manuscript).*

2. Conceptual issue: Behavior and mental representations of lonely people

The authors report a new analysis controlling for the reaction time and missed responses. Lonely participants missed fewer responses, suggesting differences in attention. But, controlling for these differences, the effect of loneliness on representational similarity persists. This is a nice analysis.

The authors' response describes new analyses that show no correspondence between MPFC representational similarity and similarity of trait evaluations. This indicates that the trait

evaluations do not correspond to the content of the MPFC representations (or that fMRI lacks the resolution required to identify the correspondences), and raises the question of what MPFC represents during the reported task. This should be reported and discussed in the manuscript.

The authors' response states that the results "cannot be explained by differences in perceptions of [...] traits." I would amend this to read "measured traits," as it is possible that the MPFC represents traits, just not the ones that were measured.

In lines 648-661 of the revised manuscript, the authors make several questionable claims that seem to spin the mismatch between trait judgments and MPFC results as a kind of robustness check. I think this is a misguided line of argument. Instead, I suggest the authors clearly state that the mismatch between the trait judgments and the MPFC representations is a shortcoming of the paper, while also clearly stating that their main arguments still stand. Social psychology has long had a problem where less than perfect results are wrongly rejected even when the big picture holds up just fine, and I can see why the authors would take a defensive pose, but it is not necessary here.

Response: *We are glad to hear you appreciate the new analyses controlling for reaction time and missed responses.*

While we appreciate the points you raise above, in line with the editor's recommendation, we have removed the sections of our manuscript in which we interpreted the null results related to the correspondence between MPFC representational similarity and similarity in the binary trait judgments. A benefit of this revision is that we no longer interpret the mismatch between trait judgments and MPFC results as a robustness check.

Also, we would like to note that we no longer say the phrase "cannot be explained by differences in perceptions of [...] traits." We also do not say "measured traits" simply because we no longer make interpretations about null findings.

3. Scope of generalization

The authors' response that celebrities were selected on the basis of popularity is simply not germane to the question of generalization scope. I would strongly encourage the authors to review Clark (1973) and other work on the stimulus-as-fixed-effect fallacy, e.g., Westfall, Nichols, & Yarkoni (2016). That said, the caveat added on line 610 is reasonable, although the authors should add that it is also not clear whether the results will generalize to other celebrities.

Response: *We have edited the caveat you reference above to also acknowledge that it is unknown "whether the results generalize to representations of all celebrities" (see page 49, lines 1113-1114 of main manuscript).*

I appreciate that the tests using celebrity categories are now identified as post hoc. But I would like to see the small number of celebrities in each category to be directly identified as a limitation in the discussion section. This would be a natural addition to the paragraph starting on line 636.

Response: *Per your recommendation, we directly identify the small number of celebrity targets as a limitation in the discussion section in the paragraph you reference above (see page 50, lines 1132-1136 of main manuscript).*

Line 642: I suggest changing "it is unclear if the MPFC findings would generalize to several celebrities" to "it is unclear if the MPFC findings would generalize to the wider population of celebrities."

Response: *We have made the edit you suggest above (see page 50, lines 1135-1136 of main manuscript).*

Line 73: I suggest changing "logical" to "reasonable."

Response: *We have made the edit you suggest above (see page 4 line 72 of main manuscript).*

4. Justification of celebrity trait ID paradigm

The authors note that the UCLA loneliness scale is well-validated, which is why they use overall loneliness rather than the specific relevant survey items. This is all well and good, but it is not germane to the concern that the jump from loneliness to celebrities is poorly justified.

Happily, the authors' reorganization of the introduction fully addresses my concern.

Response: *We appreciate the point you raise above and are glad to hear that our revised introduction has addressed your concern.*

5. Split by level of consensus

I am delighted to see the improvements to Study 2.

The authors note that in Study 1, where Justin Bieber was the only high-consensus celebrity, representing consensus continuously eliminated the effect. This is consistent with the concern that generalizing from small numbers of celebrities (in this case a single celebrity) is not appropriate. The authors note that this result may be driven by "limited variability in consensus across targets," which seems an understatement given that one category consists only of a single target, Justin Bieber. I think the paper would be stronger if this analysis were simply removed, as Study 2 now makes the same point but much more convincingly.

Relatedly, I do not find the authors' claim that "the interaction between loneliness and level of consensus was consistent across two different datasets" compelling, because the authors acknowledge that the effect in one of the datasets disappears when consensus is evaluated as a continuous variable. Why grasp at straws when the improvements to Study 2 are so strong?

Response: *Thank you for your comments on our previous draft of the manuscript that led to the improvements to Study 2 that you note. As to your suggestion that we remove the interaction result from Study 1, we appreciate your point but respectfully disagree that the paper would be improved by doing so. This is because the result from Study 1A explains why we included more celebrity targets in Study 2--so that we had an opportunity to follow-up on this result with more variability in the consensus between celebrities. In our view, reporting the result in Study 1A provides important information about the rationale behind Study 2's design and analysis.*

In line with your recommendation above, we now explicitly acknowledge in the discussion section that the small number of celebrities per category is a limitation of the study (see page 50, lines 1132-1136 of main manuscript). Relatedly, as you note above, we are transparent about the fact that the interaction is not statistically significant when modeled continuously. In our view, readers should have enough information available to them to decide for themselves whether they find this particular result convincing or not.

The authors note that Justin Bieber's shift from high-consensus in Study 1 to low-consensus in Study 2 may reflect "differences in how the target individual was generally perceived across two different points in time and/or groups of people," arguing that this "could be considered a strength of our findings." This is an interesting possibility, but it is an empirical matter and the authors provide no supporting data or analysis.

Response: *You are correct that we do not have any data that can speak to the possibility we raised. We also point out that we raised this possibility only in our response to the comments and that it does not appear in the manuscript.*

We raised this possibility in our previous response simply to point out that it may not be as strange as it seems at first glance that Justin Bieber is high consensus in one study but low

consensus in the other, and that a lack of correspondence between MPFC consensus and semantic consensus is not the only possible explanation for this ostensible inconsistency.

6. Distance-weighted average

I am happy to see that results using a distance-weighted mean without MDS are nearly identical to the reported results. I am also happy to see that the results hold without any distance-weighting at all. So why continue to report the MDS results? MDS throws information away, biasing the mean. It may not make a meaningful difference here, but it may make a difference for future studies by other authors that use this one as a template. Even the authors in their revised manuscript make no mention of MDS when justifying this analysis (line 114). Simply using the distance-weighted mean would also resolve the concern that the Gaussian kernel parameters were set arbitrarily.

Response: *It is fair to point out that in our previous response we did not justify the use of MDS. In our view, a considerable advantage of MDS is that, because it projects participants into two-dimensional space, pairwise dis/similarity in the sample can be visualized in a way that allows it to be understood more intuitively and for assumptions to be inspected. Mainly what we mean by this is that in addition to simplifying the process of identifying a single point of convergence, MDS makes it relatively simple to visually inspect whether the assumption that there is a single point of convergence is accurate or not. As mentioned in our response to the previous round of comments, it was the case in the current study that there was always a single point of convergence with participants further from this point tending to be less similar to other participants. We never observed evidence of a second, weaker point of convergence (let alone third, fourth, etc.), which increased our confidence that interpreting this point of convergence as reflecting the “zeitgeist” was appropriate. We think that being able to visualize this relationship facilitates understanding of what the analysis reflects conceptually, which may be particularly beneficial to trainees early in their research careers. We have updated our manuscript to ensure we sufficiently justify the use of MDS (see page 19, lines 430-433 of main manuscript).*

All that being said, we appreciate the concern you raise regarding the loss of information. Therefore, in addition to being clearer about what we see as the benefits of MDS, we also make sure to acknowledge this drawback in the revised manuscript (see page 19, lines 426-427 of main manuscript). Further, we now report the results of all relevant analyses using the distance-weighted mean without MDS alongside our “point of convergence” approach using MDS and the unweighted mean. Any researchers using our study as a template will now know that it is advisable to ensure their results hold using the distance-weighted average without MDS.

In their response, the authors argue their approach would better capture multi-modal distributions of representations (i.e., more than one consensus). But this does not match the reported methods, as it would require identifying at least two centroids using at least two Gaussian kernels and a further process to optimize such a model. In any case, while a Gaussian mixture model approach is justifiable, it also would not require MDS.

Response: *The purpose of the hypothetical example we described was to identify a situation in which the “point of convergence” approach and distance-weighted mean approach would no longer be virtually equivalent. We were not, however, suggesting that one would need to identify both centroids if a weaker consensus was present. The concept of the “zeitgeist” as the most common shared perceptions in a given culture should be represented as a single point of convergence across people. The hypothetical secondary, weaker point of convergence could conceivably represent a relatively commonly held countercultural view that, despite being somewhat common, would still not represent the zeitgeist, i.e., the most dominant cultural view. Participants’ weight in the weighted average would still be based only on their proximity to the single point of greatest convergence across participants. You could imagine the participants that are dissimilar to the consensus as falling on an equidistant orbit encircling the point of greatest convergence, with a single participant placed on the top of this orbit and a group of several participants arranged along the bottom of this orbit. All of these participants would be given the same weight in the “point of convergence” approach because they are all equidistant from this point. But in the distance-weighted mean approach the participants forming the second, weaker cluster along the bottom of this orbit would be given greater weight than the solitary one at the top.*

A situation like the one described above might not have the same implications for loneliness as the one in which there is a single, strong point of convergence, as a second point of weaker convergence might still represent an opportunity for community with respect to one’s perspectives (even if outside the mainstream). This is a possibility that future work will have to explore as we never encountered this situation in the present study.

19th Mar 24

Dear Dr Broom,

Your manuscript titled "Keeping up with others' perceptions of the Kardashians: Lonely individuals' neural representations and language use do not reflect the zeitgeist" has now been reviewed by the editorial team. I am delighted to say that we are happy, in principle, to publish a suitably revised version in *Communications Psychology* under the open access CC BY license (Creative Commons Attribution v4.0 International License).

We therefore invite you to revise your paper one last time to address the remaining concerns of our reviewers and a list of editorial requests. At the same time we ask that you edit your manuscript to comply with our format requirements and to maximise the accessibility and therefore the impact of your work.

EDITORIAL REQUESTS:

SUBMISSION INFORMATION:

OPEN ACCESS:

Communications Psychology is a fully open access journal. Articles are made freely accessible on publication under a CC BY license (Creative Commons Attribution 4.0 International License). This license allows maximum dissemination and re-use of open access materials and is preferred by many research funding bodies.

For further information about article processing charges, open access funding, and advice and support from Nature Research, please visit <https://www.nature.com/commspsychol/article-processing-charges>

At acceptance, you will be provided with instructions for completing this CC BY license on behalf of all authors. This grants us the necessary permissions to publish your paper. Additionally, you will be asked to declare that all required third party permissions have been obtained, and to provide billing information in order to pay the article-processing charge (APC).

* **TRANSPARENT PEER REVIEW:** *Communications Psychology* uses a transparent peer review system. On author request, confidential information and data can be removed from the published reviewer reports and rebuttal letters prior to publication. If you are concerned about the release of confidential data, please let us know specifically what information you would like to have removed. Please note that we cannot incorporate redactions for any other reasons.

* CODE AVAILABILITY: All Communications Psychology manuscripts must include a section titled "Code Availability" at the end of the methods section. We require that the custom analysis code supporting your conclusions is made available in a publicly accessible repository at this stage; please choose a repository that generates a digital object identifier (DOI) for the code; the link to the repository and the DOI must be included in the Code Availability statement. Publication as Supplementary Information will not suffice.

* DATA AVAILABILITY:

[Link redacted]

Best regards,

Jennifer Bellingtier

Jennifer Bellingtier, PhD
Senior Editor
Communications Psychology